# Sympathetic Regulation of Hematopoiesis and the Mobilization of Inflammatory Cells in ICR Mice with Traumatic Brain Injury: A Novel Approach to Targeting Neuroinflammation and Degenerative Processes

**DOI:** 10.3390/biomedicines13123080

**Published:** 2025-12-13

**Authors:** Natalia Ermakova, Victoria Skurikhina, Edgar Pan, Mariia Zhukova, Irina Zharkikh, Valentina Pan, Alexander Dygai

**Affiliations:** Institute of General Pathology and Pathophysiology, Moscow 125315, Russia; nejela@mail.ru (N.E.); vskurikhina@mail.ru (V.S.); mashazyk@gmail.com (M.Z.); terekhina@rambler.ru (I.Z.); valentinaglazacheva@gmail.com (V.P.); amdygay@gmail.com (A.D.)

**Keywords:** traumatic brain injury, neuroinflammation, degenerative processes, reserpine, hematopoiesis, mobilization, inflammatory cells

## Abstract

**Background/Objectives**: Neuroinflammation is a leading factor in secondary brain damage following a traumatic brain injury (TBI). Existing therapeutic approaches have limited efficacy against neuroinflammation. The bone marrow, the primary hematopoietic organ, is also a source of inflammatory cells. We propose that targeting the sympathetic regulation of inflammatory cell mobilization could reduce neuroinflammation after TBI. **Methods**: In ICR mice, we investigated the immune cell response in the blood, bone marrow, motor cortex, and the subventricular zone after TBI modeling and treatment with the sympatholytic agent reserpine. **Results**: TBI induced neutrophilia and lymphocytosis in the peripheral blood, activated hematopoiesis in the bone marrow, and triggered neuroinflammation and degenerative changes in the cerebral cortex (CC) and the subventricular zone (SVZ) of mice. Reserpine reduced leukocytosis in the blood and hematopoietic activity in the bone marrow of mice with TBI compared to untreated TBI mice. Furthermore, reserpine decreased neutrophilic and lymphocytic infiltration, as well as the number of Iba1^+^ microglial cells, including M1-polarized microglia, Caspase-3^+^ cells, and cells expressing myeloperoxidase (MPO) in the CC and SVZ of treated mice. The activity of degenerative processes was also reduced. Additionally, reserpine reduced the number of M2-polarized microglial cells in the SVZ. **Conclusions**: The sympatholytic drug reserpine may hold promise for the development of a novel approach to treating neuroinflammation and degeneration following a TBI. This is based on its ability to reduce hematopoiesis and mobilize inflammatory cells from the bone marrow into the bloodstream.

## 1. Introduction

Traumatic brain injury (TBI) remains a significant challenge in modern healthcare [1]. Despite a global trend toward reduced trauma incidence, approximately 64–74 million cases of moderate or severe TBI are reported annually, accounting for 69% of all TBI cases [2,3]. The mortality rate averages 30%, and 50% of patients with severe TBI acquire disabilities within one year [4]. Even six months after mild TBI, half of the hospitalized patients fail to achieve full recovery [2]. The long-term effects of TBI encompass not only persistent physical and cognitive impairments but also an increased risk of developing neurodegenerative diseases, such as Alzheimer’s and Parkinson’s disease [5].

Surgical treatment of TBI is aimed at reducing intracranial hypertension, evacuating hematomas and areas of contused tissue. Conservative therapy includes measures to support vital functions (e.g., mechanical ventilation, nutritional support) and prevent secondary brain injury [6]. Despite numerous preclinical and clinical studies, many potential therapeutic strategies have failed to demonstrate the expected efficacy in multicenter randomized controlled trials. For instance, the effectiveness of the neurosteroid allopregnanolone and the antibiotic minocycline as neuroprotective agent, as well as progesterone in the treatment of moderate and severe TBI was not confirmed (NCT01673828) [7,8]. Amantadine accelerated the functional recovery of patients with long-term impaired consciousness after TBI; however, the effect diminished after drug discontinuation. Positive results from using a ghrelin agonist, OXE-103, in brain injury were recorded based only on the Post-Concussion Symptom Scale (PCSS) about concussion symptoms (NCT0455834) [9]. The nitric oxide synthase inhibitor VAS203, improved clinical outcome of patients with TBI in a phase IIa trial. However, at the highest doses, VAS203 was associated with acute kidney injury [10]. Cell therapy is a promising option for the TBI treatment. Several clinical studies are currently evaluating the efficacy and safety of different types of cell therapy, including the infusion of autologous bone marrow mononuclear cells and mesenchymal stem cells, in patients with mechanical brain injury (NCT01851083, NCT01575470, NCT02525432, NCT02028104, NCT00254722, NCT06146062) [11,12,13]. Research is also assessing the effectiveness of beta-adrenergic blockers for correcting sympathetic hyperactivation (NCT06569212) and the inhibition of the complement system following brain injury [14]. Yet, the results of these studies have not yet been published, or the trials are still ongoing.

In searching a new approach for neuroinflammation treatment in TBI, we have focused on the crucial process as the inflammatory cell mobilization from the bone marrow into the blood and their subsequent migration to damaged tissue [15,16]. We hypothesize that it is possible to modulate neuroinflammation after mechanical trauma by regulating the immune cells mobilization from the bone marrow. In our search for ways to influence this process, we looked at publications discussing the relationship between the sympathetic nervous system (SNS) and immune cells [17,18]. We believe that the mobilization of immune cells from the bone marrow into the bloodstream can be modulated using a sympatholytic agent. In the previous studies, we successfully used the sympatholytic drug reserpine to reduce inflammation in experimental models such as diabetes mellitus and pulmonary fibrosis [15,16].

The main aim of this study was to investigate the role hematopoiesis in the development of neuroinflammation and to assess reserpine anti-inflammatory effects in male ICR mice with TBI.

## 2. Materials and Methods

### 2.1. Animals

Experiments were conducted on 136 outbred male ICR mice aged 12–14 weeks, obtained from the department of Experimental Animal Clinic, Institute of General Pathology and Pathophysiology (Moscow, Russia). All manipulations with animals were maintained in accordance with the European Convention for the Protection of Vertebrates (Strasbourg, 1986) and the Principles of Good Laboratory Practice (OECD, ENV/MC/CUEM (98)17, 1997). The study was approved by the Ethics Committee of the Institute of General Pathology and Pathophysiology (Protocol No. 2 of 12 March 2024. The mice were housed five to six mice per cage with a 12 h light/dark cycle. Throughout the experimental period, the mice had free access to drinking water and standard rodent chow. The animals were sacrificed by CO_2_ overdose.

### 2.2. Experimental Groups and Study Design

The animals were divided into three groups: group 1—intact mice (n = 56); group 2—mice with TBI (n = 56); group 3—mice with TBI treated with reserpine (n = 24). At the time points shown in Table 1 samples of peripheral blood from the tail vein, bone marrow, and brain were collected. The distribution of animals, depending on the type of sample and the sampling time, is shown in Appendix A, Table A1.

The white blood cell (WBC) count and the leukocyte formula were assessed in the blood samples. The content of various karyocyte forms, hematopoietic precursors, and hematopoietic islands was evaluated in the bone marrow. Severity of tissue damage, neuroinflammation, degenerative changes, and the number of cells expressing Iba1, iNOS, Caspase-3, and CD206 in the cerebral cortex (CC) and subventricular zone (SVZ) of the right cerebral hemisphere were examined.

At the first stage of the study, we assessed the response of the blood system and the mechanisms of its development in mice following modeled TBI. At the second stage, we evaluated the effects of reserpine on the blood system. Finally, at the third stage, we investigated the influence of reserpine on damage, neuroinflammation, and degenerative changes in the CC and the SVZ of the right hemisphere.

### 2.3. Traumatic Brain Injury Model

Experimental TBI was induced using a focal impact from a free-falling weight on the skull bones under inhalation anesthesia (isoflurane, Sigma-Aldrich, Saint Louis, USA) [2,3]. Using a stereotaxic apparatus, a blunt metal impactor tip with a diameter of 5 mm and a mass of 36 g was centered and positioned over the point where the coronal and sagittal sutures meet (the bregma point) [19,20]. The impactor tip was raised to a height of 14 cm and dropped along a guide tube. Verification of the injury at the target location was confirmed during postmortem examination. Animals were excluded from the experiment if there was a ricochet of the impactor, skull fracture, or if the impact site was displaced more than 2 mm from the bregma. The day of TBI induction was designated as Day 0 of the experiment.

### 2.4. Reserpine

Reserpine (Sigma-Aldrich, USA)–methyl reserpate 3,4,5-trimethoxybenzoate, an indole alkaloid derived from Rauvolfia serpentina. The primary pharmacological property of reserpine is its sympatholytic effect, which is mediated by the accelerated release of catecholamines from presynaptic nerve terminals and their subsequent inactivation by monoamine oxidase (MAO).

### 2.5. Administration of Reserpine

This study investigated the effects of low-dose reserpine (0.1–1.0 mg/kg) on hematopoiesis and neuroinflammation. Reserpine was dissolved in physiological saline and administered intraperitoneally in a volume of 0.1 mL on days 3, 4, 5, 6, and 7 after injury. The first reserpine injection was administered at a dose of 1.0 mg/kg, followed by subsequent injections at a dose of 0.1 mg/kg. TBI and intact mice were administered intraperitoneally an equivalent volume of solvent (0.1 mL). The doses and regimens for administering reserpine were selected based on our previous research, which demonstrated the anti-inflammatory effects of reserpine in various conditions [16,21,22], as well as a study demonstrating the development of degenerative changes on 3rd day after TBI [23].

### 2.6. Peripheral Blood Leukocytes

In peripheral blood, we examined the total number of white blood cells (WBCs) and the count of morphologically identifiable white blood cells types [24].

### 2.7. Bone Marrow Cells

Bone marrow was isolated from at 5 h, 1 day, 3 days, 7 days, and 21 days. In the bone marrow, we assessed the total number of cells and the count of morphologically identifiable cell types [24].

### 2.8. Bone Marrow Hematopoietic Islands

Bone marrow was isolated from mouse femurs on days 7 and 21. We analyzed the quantitative and qualitative composition of hematopoietic islands (HIs) in the bone marrow [24].

### 2.9. Bone Marrow Hematopoietic Precursors

Bone marrow was isolated from mouse femurs using 0.5 mL of preparation medium (95% RPMI-1640 and 5% FBS), resuspended, and centrifuged. To isolate the non-adherent mononuclear cell fraction, 2 × 10^6^ cells/mL were incubated for 45 min in a medium containing 90% RPMI-1640 and 10% FBS in 90 mm Petri dishes at 37 °C, 5% CO_2_, 100% humidity. To assess the colony-forming activity of granulocytes, colony-forming units granulocytes (CFU-G) were calculated, the fraction of nonadherent mononuclear cells in the amount of 0.2 × 10^6^ cells/mL was cultured in RPMI-1640 medium (Servicebio, Wuhan, China) supplemented with 30% methylcellulose (CDH, New Delhi, India), 10% fetal bovine serum (FBS) (Servicebio, Wuhan, China), 280 mg/mL L-glutamine (Servicebio, China), 0.4 × 10^−5^ M 2-mercaptoethanol (Servicebio, Wuhan, China) and 5 ng/mL recombinant granulocyte colony-stimulating factor (Tevagrastim, “Teva Pharmaceutical Industries, Ltd.”, Kfar Saba, Israel) at 37 °C, 5% CO_2_, 100%. Granulocyte colonies were counted on the 7th day [25].

### 2.10. Histological Examination

Brain was isolated from mice on days 3, 7, and 21 after TBI. Histological sections were prepared and stained with hematoxylin and eosin using standard protocols [23]. During the histological examination, we focused on the CC at the site of injury and the SVZ of the right hemisphere of the brain. We used a semi-quantitative scoring system to evaluate tissue and cellular edema, vascular structural disorders, the severity of inflammatory infiltration, the presence of hemorrhages, the fullness of blood vessels, the presence of neurons with hypochromic and hyperchromic nuclei, pycnotic nuclei, vacuolated cells, and dividing neuroglial cells on micrographs of the prepared tissue [23].

### 2.11. Immunohistochemical Examination

Immunohistochemical (IHC) analysis of the CC and SVZ in the right cerebral hemisphere was conducted in the damaged area (from +0.50 to −0.10 mm relative to the bregma). For this purpose, the brains were extracted and placed in 10% formaldehyde solution, followed by standard cryo-preparation using sucrose solutions of increasing concentration in formaldehyde-based buffer. The tissue was frozen in liquid nitrogen vapor and stored at −80 °C. Cryosections with a thickness of 10 microns were prepared using a cryostat (MCM-3500, MT Point, Saint-Petersburg, Russia). Specific cellular markers were identified using recombinant mouse monoclonal antibodies against ionized calcium-binding adapter molecule 1 (Iba1) (Cat.No GB15105, Mouse Anti-Iba1, Servicebio Technology, Wuhan, China), rabbit polyclonal antibodies against the inducible nitric oxide synthase (iNOS) (Cat.No GB11119, Rabbit Anti-iNOS, Servicebio Technology, Wuhan, China), rabbit polyclonal antibodies against the macrophage mannose receptor (Mannose Receptor/CD206) (Cat.No GB113497, Rabbit Anti-Mannose Receptor/CD206, Servicebio Technology, Wuhan, China), rabbit polyclonal antibodies against caspase-3 (Caspase-3) (Cat.No GB11532, Rabbit Anti-Cleaved-Caspase-3, Servicebio Technology, Wuhan, China), rat monoclonal antibody against Myeloperoxidase (MPO) (Cat.No ab300650, Rat Anti-Myeloperoxidase [EPR20257], Abcam, Massachusetts, USA). Antibodies conjugated with FSD™ 488 (Cat.No RSA1245, Goat anti-rabbit IgG, FSD™ 488) and FSD™ 594 (Cat.No RSA1195, Goat anti- mouse IgG, FSD™ 594) (both from BioActs, Incheon, in the Republic of Korea), and also antibody conjugated with DyLight^®^ 488 (Cat.No ab102260, Donkey Anti-Rat IgG H&L (DyLight^®^ 488) pre-adsorbed, Abcam, Massachusetts, USA) were used as secondary antibodies. The dilution of the primary antibodies was 1:1000 and the dilution of the secondary antibodies was 1:2000. After staining, the slices were covered with a mounting medium containing DAPI Mounting Medium With DAPI—Aqueous, Fluoroshield, Cat.No ab104139, Abcam, USA).

Micrographs of brain samples stained with IHC were obtained using a direct fluorescence microscope Olympus BX51 (Olympus Optical Co., Tokyo, Japan) equipped with an Olympus XM10 camera (Olympus Optical Co., Japan). Image analysis was performed using ImageJ software (NIH) (ver. 1.54p). For each staining, microphotographs of 5 brain sections were taken. Subsequently, in 3 fields of view (200 × 200 μm area each), we quantified the total number of nuclei and the percentage of cells expressing the microglial marker Iba1 in combination with either iNOS, CD206, or Caspase-3.

### 2.12. Statistical Analysis

Statistical analysis was performed using SPSS (version 15.0, SPSS Inc., Chicago, IL, USA). Data were analyzed and presented as M ± m. The normality of data distribution was assessed using the Shapiro–Wilk test. Statistical significance was evaluated by Student’s *t*-test (for parametric data), or Mann–Whitney test (for nonparametric data) when appropriate. A *p*-value of less than 0.05 (by two-tailed testing) was considered an indicator of statistical significance. For multiple comparisons, the Bonferroni correction was applied to adjust the *p*-value (*p* < 0.017).

## 3. Results

### 3.1. TBI-Induced Changes in the Blood System

#### 3.1.1. Leukocytosis in Peripheral Blood

TBI modeling caused an increase in the number of white blood cells in mice starting from 15 min and lasting up to 21 days, with the maximum occurring from 1 h to 4 days after the injury. The leukocytosis at 1–5 h was primarily driven by an increase in segmented neutrophils. Starting from 5 h, the contribution of lymphocytes to the leukocytosis became more pronounced (Figure 1). Eosinophils, monocytes, and plasmocytes contributed less to the overall leukocytosis compared to neutrophils and lymphocytes (Table 2).

#### 3.1.2. Hematopoietic Bone Marrow Hyperplasia

The bone marrow serves as a source of inflammatory cells. The mobilization of inflammatory cells from the bone marrow into the blood represents a key mechanism of inflammation. In this context, we investigated inflammatory cells in the blood and bone marrow following a TBI. Our findings suggest that the response of bone marrow cells to TBI is complex. Specifically, the total number of karyocytes in the bone marrow decreased relative to the intact control at 1 h, 1 day, and 3 day post-injury. In contrast, the total number of karyocytes increased on days 7, 14, and 21. At the time points of the most pronounced increase in total number of karyocytes—days 7 and 21—bone marrow specimens were examined. The results revealed an increase in the number of myeloid and lymphoid cells, monocytes, plasmacytes, and erythroid cells in the bone marrow. Concurrently, the mitotic activity of myeloid cells was enhanced (Figure 2).

#### 3.1.3. Enhanced Colony-Forming Activity of Bone Marrow Hematopoietic Precursors

The response of bone marrow hematopoiesis to various extreme factors largely depends on the activity of hematopoietic precursors that give rise to blood cells [25]. We observed increased granulocyte precursor activity post-TBI compared to intact control. This enhancement was detected at 1 and 5 h, and at 1 and 3 days after injury. Furthermore, CFU-G cellularity in the TBI group was 26–30% higher than in the intact group (Table 3).

#### 3.1.4. Formation of Granulocytopoiesis Foci in the Bone Marrow

The maturation of myelokaryocytes occurs within hematopoietic islands (HIs) [24]. TBI modeling increased the number of macrophage-negative HIs in the bone marrow of mice (days 7 and 21) (Figure 3). The study of cytological preparations revealed additional formation of HIs not associated with macrophage elements, which contained promyelocytes, myelocytes, metamyelocytes, and neutrophils (days 7 and 21) (Figure 3). The numbers of erythroid HIs and erythro-granulocytic HIs containing both myeloid and erythroid cells remained unchanged throughout all observation time points.

### 3.2. Histological and Immunohistochemical Studies of the Cerebral Cortex and the Subventricular Zone After TBI

#### 3.2.1. Tissue Damage, Neuroinflammation, and Degenerative Changes

Analysis of mouse brain sections stained with hematoxylin and eosin revealed inflammatory infiltration, comprising lymphocytes, cells of the mononuclear phagocyte system, and neutrophils, along with edema, signs of stasis, vascular congestion, and hemorrhages against a background of compromised vascular wall integrity in CC and SVZ of the right hemisphere at the injury sites (days 3, 7, and 21) (Figure 4). The most pronounced severity of this complex of pathological alterations was observed on days 3 and 7 post-injury (Figure 4). Furthermore, signs of degenerative changes were detected in these same areas, namely glial cell mitoses, vacuolization of glial and neural cell nuclei. Neurons with hyperchromatic and pyknotic nuclei were also identified. A glial scar was forming in the damaged area. These features were most clearly evident on day 21 (Figure 4).

#### 3.2.2. Changes in the Number of Cells Expressing Iba1, iNOS, Caspase-3, and CD206

IHC analysis of the CC and SVZ of the right brain hemisphere was performed at the peak of neuroinflammation (day 7 post-injury). TBI reduced the total number of cells in both brain regions in TBI mice compared to the intact control (Figure 5 and Figure 6). Nevertheless, the number of Iba1^+^ microglial cells increased, particularly that of amoeboid Iba1^+^ cells, which are defined as actively migrating pro-inflammatory M1-polarized cells (Figure 5 and Figure 6) [26]. Additionally, we assessed the expression of iNOS, Caspase-3, CD206 and MPO in Iba1^+^ cells. As shown in Figure 5 and Figure 6, TBI increased the number of Iba1^+^iNOS^+^ cells (M1-polarized microglial cells) in the studied brain structures; however, these changes were not statistically significant.

Expression of the apoptotic marker Caspase-3 increased in Iba1^+^ cells of the SVZ. In contrast, the number of Iba1^+^Caspase-3^+^ cells (microglial cells undergoing apoptosis) in CC of the TBI group did not differ from that in the intact control group (Figure 7).

The results for the CD206 marker were of particular interest. TBI reduced the number of Iba1^+^CD206^+^ cells (M2-polarized microglia) in the CC of TBI mice compared to the intact control, but did not affect the number of Iba1^+^CD206^+^ cells in the SVZ (Figure 8). Furthermore, TBI led to an increase in the number of cells expressing MPO in the SVZ, whereas their number in the CC was unchanged compared to intact animals (Figure 9).

### 3.3. Effect of Reserpine on the Blood System in TBI

#### 3.3.1. Reserpine Reduces the White Blood Cell Count in Blood

Reserpine prevented the development of leukocytosis in TBI mice on days 3–7 and 14 (Figure 10). This effect was associated with a reduction in the numbers of neutrophils (days 4, 6, 7, 14, 21), eosinophils (days 5, 6, 21), plasmocytes (day 21), and lymphocytes (days 3–7, 14, 21) (Table 4).

#### 3.3.2. Reserpine Has a Complex Effect on Bone Marrow Cells

The response of bone marrow cells from TBI mice treated with reserpine was ambiguous. Specifically, the numbers of myelocytes, metamyelocytes, neutrophils, lymphocytes, monocytes, and plasma cells increased in reserpine-treated TBI mice at day 7 compared with the intact group. This led to an elevation in the total number of karyocytes in reserpine-treated mice compared to TBI mice (Figure 11).

#### 3.3.3. Reserpine Does Not Affect the Clonal Activity of Bone Marrow Hematopoietic Precursors

Reserpine did not affect the clonal activity of granulocyte precursors in the bone marrow of mice with TBI (Table 3). CFU-G cellularity in TBI mice treated with reserpine did not differ significantly from that in the untreated animals.

#### 3.3.4. Reserpine Exerts a Complex Effect on Hematopoietic Islands

Reserpine increased the number of macrophage-negative HIs in the bone marrow of TBI mice on day 7 compared to the intact mice (Figure 12). Furthermore, the formation of HI not associated with macrophage elements composed of myeloid cells (granulocyte HI) was observed in treated animals. In contrast, by day 21, reserpine reduced the numbers of macrophage-negative His compared to the untreated TBI mice (Figure 12).

### 3.4. Effect of Reserpine on Brain Morphology After TBI

#### 3.4.1. Reserpine Reduces Neuroinflammation and Degenerative Changes

Hematoxylin and eosin staining of brain sections showed that reserpine did not significantly affect tissue damage in the CC of TBI mice compared to the untreated mice on days 7 and 21 (Figure 4). However, in both the CC and SVZ, reserpine reduced edema and neuroinflammation, as well as stasis and degenerative phenomena at the tissue and cellular levels. The disruption of the vascular network, astrogliosis, and glial scar formation observed in the reserpine-treated group were less pronounced than in the TBI group.

#### 3.4.2. Reserpine Reduces the Number of Cells Expressing Iba1, iNOS, CD206, Caspase-3, and MPO

Reserpine did not significantly affect the total number of cells in the MC and SVZ in the reserpine-treated group compared to the TBI group on day 7 of the experiment (Figure 5 and Figure 6). However, in the studied brain regions of the reserpine-treated group, the number of Iba1^+^ microglial cells were significantly reduced. A decrease in the number of cells with amoeboid morphology, M1-polarized microglial cells (Iba1^+^iNOS^+^), and Iba1^+^Caspase-3^+^ cells in the total Iba1^+^ cell population was observed (Figure 5, Figure 6 and Figure 7).

Reserpine administration did not affect the content of cells with co-localization of Iba1 and CD206 markers in the microglia of the CC in the reserpine-treated group relative to the TBI group, while their number decreased in the SVZ (Figure 8). Furthermore, reserpine significantly reduced the number of cells expressing MPO in both the CC and SVZ (Figure 9).

## 4. Discussion

The main targets for countering inflammation are chemokines and inflammatory cells that infiltrate damaged tissue [27]. At the same time, the bone marrow, which is not only the main organ for hematopoiesis but also a source of immune cells, is largely ignored in the search for new therapeutic targets. It is within the bone marrow that the division and differentiation of hematopoietic stem cells and hematopoietic precursors occur, along with cellular maturation, occurs. Furthermore, the mobilization of immune cells from the bone marrow into the bloodstream represents a critical step in the formation of inflammatory responses and immune reactions. Thus, the bone marrow represents a potential therapeutic target for modulating inflammation.

Considering the connection between the immune system and the SNS, we investigated the possibility of modulating neuroinflammation following a TBI by targeting the sympathetic regulation of bone marrow hematopoiesis [28]. Our approach focused on hematopoiesis and the inflammatory cell mobilization from the bone marrow into the bloodstream.

Current data on changes in leukocyte counts following a TBI are conflicting. While some authors describe TBI-induced immunosuppression, others report the development of leukocytosis [29,30,31]. Furthermore, there is still no consensus regarding changes in neutrophil and lymphocyte levels in the blood in response to mechanical trauma. The first stage of our study aimed to address these questions. We observed leukocytosis in TBI mice throughout the entire observation period, starting from 15 min and continuing until the end of the study (Figure 1). While the leukocytosis observed within the first 5 h post-injury was predominantly driven by an increase in neutrophil count, the elevated WBC count at later time points was primarily associated with lymphocyte mobilization into the bloodstream (Figure 1). Our data are largely consistent with previously published findings by other researchers, who have reported that an early predominance of neutrophils after TBI has been observed both in experimental TBI and clinical trials [29,31,32].

When comparing these data with the results of the bone marrow investigation, it became apparent that during the period of most active lymphocytes and neutrophils mobilization into the blood (from 5 h to 3 days), a decrease the cell count in the bone marrow was observed. We interpreted this as a consequence of the mobilization of bone marrow cells into the bloodstream, which is a standard response to the influence of various pathological factors. Restoration of bone marrow cellularity had taken place from day 7 through day 21. A detailed analysis of myelograms revealed that the hyperplasia developing during this period was primarily driven by a marked increase in immature myeloid cells (myelocytes, metamyelocytes), as well as neutrophils and lymphocytes.

Taken together, these data and the brain response to injury suggest that inflammatory infiltration in the CC and SVZ after TBI may be largely driven by neutrophils, lymphocytes, and monocytes/macrophages recruited from the bone marrow into the blood. This hypothesis is further supported by results from several studies indicating that leukocytes can migrate from the skull bone marrow into the brain following a TBI [33,34].

As is known, the maturation of bone marrow karyocytes occurs within HIs [24]. TBI modeling caused changes in the structural and functional organization of the mice bone marrow on days 7 and 21. In TBI mice, there was formation of additional HIs not associated with macrophages, which contained promyelocytes, myelocytes, metamyelocytes, and neutrophils (Figure 2). On the other hand, bone marrow has been considered as a source of poorly differentiated cells [12,35]. When extreme factors of various kinds act on the body, the activity of bone marrow hematopoiesis largely depends on hematopoietic precursors [25]. The hematopoietic precursors with the Lin^−^Sca-1⁺c-kit⁺ phenotype that we isolated from the bone marrow of untreated TBI mice demonstrated high colony-forming activity of the granulocytic type (at 1 and 5 h, 1 and 3 days) in vitro compared to the intact control (Table 2). These results are consistent with existing data showing enhanced activity of hematopoietic precursors following trauma (myeloid lineage differentiation) [36].

Thus, following a TBI, the activation of bone marrow hematopoiesis is largely driven by hematopoietic precursors and hematopoietic islands. These mechanisms maintain a high concentration of immune cells and blood cells in the brain over an extended period, while mobilization and migration processes ensure their delivery to the damaged brain. From this perspective, we propose considering the bone marrow and the process of mobilization as promising therapeutic targets for modulating neuroinflammation in TBI.

The SNS is a crucial regulator of bone marrow niches [37]. Evidence indicates that following a TBI, the sympathetic component of the nervous system becomes activated, characterized by norepinephrine hypersecretion and enhanced adrenergic regulation of bone marrow hematopoiesis, which ultimately leads to an increase in the number of myeloid precursors in the bone marrow [38]. Furthermore, clinical observations have demonstrated the efficacy of the β-adrenergic blocker propranolol in reducing sympathetic overactivation following a TBI [39,40]. When sympathetic impulse is blocked via a ligand–receptor mechanism, it does not significantly reduce norepinephrine levels in the SNS. Under conditions of extreme stress or disease, this sustained elevation of norepinephrine may have negative consequences due to norepinephrine-induced damage to cells and tissues. Therefore, administration of the sympatholytic agent reserpine, which depletes catecholamine stores, may represent a safer and more effective approach to reducing the severity of the hematopoietic response after TBI compared to the use of adrenergic blockers.

In the present study, reserpine was administered in low doses to reduce its central nervous system side effects. Reserpine doses were selected based on our previous studies demonstrating its activity, including anti-inflammatory, in various mice models (diabetes mellitus, idiopathic pulmonary fibrosis, myelosuppression) at similar doses [15,16,21,41], as well as studies related to the effects of reserpine in various doses on the central nervous system [42,43]. The reserpine treatment was administered over a 5-day course starting on day 3 post-injury, a period when neuroinflammation is pronounced and continues to progress (Figure 1 and Figure 4, Table 2). The treatment resulted in a reduction in the numbers of all white blood cell forms in the blood of animals with TBI, observed both 2 h after the first injection of the drug and on days 4–7, 14, and 21. In the bone marrow, the numbers of myeloid cells, plasmocytes, and monocytes were increased on day 7, while the numbers of myeloid and lymphoid cells were decreased on day 21. Reserpine did not affect the clonal activity of hematopoietic precursors but suppressed the formation of hematopoietic islands responsible for myeloid cell maturation (day 21) (Figure 12). Among these hematological effects of reserpine, the reduction in inflammatory infiltration, edema, and stasis in the CC and SVZ of the right brain hemisphere in ICR mice with TBI on days 7 and 21 is particularly noteworthy (Figure 4). Some signs of degenerative changes, such as vacuolization of astroglial and neuronal cells, and cells with hyperchromatic and pyknotic nuclei, were only occasionally observed.

These results led us to hypothesize that the sympathetic nervous system exerts a regulatory influence on bone marrow hematopoiesis and the mobilization and migration of inflammatory cells in mice following a TBI. Administration of low-dose reserpine significantly suppressed these processes, thereby mitigating neuroinflammation and degenerative processes in the CC and SVZ of the injured right brain hemisphere.

The patterns of changes in the traumatized brain are largely established. For instance, TBI leads to increased proliferation of microglial cells and astrocytes, accompanied by microglial polarization [44]. Morphologically, microglial activation was manifested by their acquisition of an amoeboid shape [26]. Resident M1 cells with a pro-inflammatory immunophenotype predominated over M2 cells, which play a crucial role in clearing tissue debris during earlier stages [45]. Some authors have reported a decrease in the number of Caspase-3^+^ cells in the injured brain, which may be attributed to the predominance of cell proliferation over apoptosis [46]. The results of the present immunohistochemical study provide new insights into the cerebral pathology of TBI. Specifically, on day 7 post-injury, we observed an increased content of Iba1^+^ cells in the CC and SVZ, including amoeboid Iba1^+^ cells, which are defined as actively migrating pro-inflammatory M1-polarized cells (Figure 5 and Figure 6). iNOS serves as a key mediator in the development of inflammatory processes [47,48]. In our study, TBI increased the number of Iba1^+^iNOS^+^ cells in both brain structures (Figure 5 and Figure 6). This indicates the involvement of pro-inflammatory M1-polarized cells in neuroinflammation.

Of particular note is the finding that the number of Iba1^+^ cells expressing Caspase-3 increases in the SVZ, which can be attributed to the induction of apoptosis in Iba1^+^ microglial cells. In contrast, the content of Iba1^+^Caspase-3^+^ cells in the CC remains unchanged. This may indicate that Iba1^+^ microglial cells in the SVZ are more susceptible to mechanical damage than those in the CC. Neutrophils are the primary sources of MPO. By producing MPO, they can exacerbate damage to the blood–brain barrier and contribute to secondary brain injury [49,50]. In the SVZ of untreated TBI mice, we observed an increase in the number of Iba1^+^MPO^+^ cells compared to the intact control, whereas the number of Iba1^+^MPO^+^ cells in the CC remained unchanged (Figure 9).

Particularly noteworthy are the results of CD206 expression, a marker of anti-inflammatory M2 microglial cells. As evident from Figure 8, TBI reduced the number of CD206^+^ cells in the CC of untreated TBI mice compared to the intact control, while leaving the number of CD206^+^ cells in the SVZ unaffected.

Based on the hypothesis that the SNS exerts regulatory control over resident microglial inflammatory cells, we examined the expression of Iba1, iNOS, CD206, Caspase-3, and MPO in cells of the CC and SVZ in mice with TBI following reserpine administration. Our analysis revealed that the most significant changes involved a reduction in the number of Iba1^+^ cells in both studied regions. This included a decrease in the subpopulation of Iba1^+^ cells with amoeboid morphology, as well as a reduction in M1-polarized microglial cells and Iba1^+^ cells exhibiting co-localization with Caspase-3 and MPO (Figure 5, Figure 6, Figure 7, Figure 8 and Figure 9). Furthermore, investigation of CD206 and Iba1 co-localization demonstrated that reserpine reduced the representation of M2-polarized microglial cells in the SVZ, while having no effect on their population in the CC.

The reduction in the number of resident proinflammatory M1- and anti-inflammatory M2-polarized microglial cells in the injured brain that we observed is not a controversial treatment outcome. On the contrary, the phenomenon fits within the logic of reserpine’s action. By changing the systemic balance of catecholamines [51,52,53], the sympatholytic drug indirectly affected the activity of both proinflammatory cells and anti-inflammatory microglial cells via α- and β-adrenergic receptors [54,55,56,57,58,59]. Another potential mechanism of the anti-inflammatory effect of reserpine is its inhibition of soluble epoxy-hydroxylase (sEH) [60]. sEH inhibition reduces the NF-κB signaling pathway activation in immune cells, leading to a decrease in the production of proinflammatory cytokines such as TNF-α [61]. On the other hand, a decrease in sEH activity was accompanied by an increase in the production of brain-derived neurotrophic factor (BDNF) by astrocytes and vascular endothelial growth factor (VEGF) by neurons, which have a neuroprotective and regenerative effect [62]. It is possible that reserpine additionally reduces the expression of iNOS by M1-polarized microglia [57,58]. The proposed mechanism of the anti-inflammatory activity of reserpine in TBI is shown in Figure 13.

Our study has certain limitations, primarily the inherent constraint of animal models to fully recapitulate the entire spectrum of human pathology following a TBI, coupled with a relatively short observation period. Furthermore, the lack of a control group receiving only reserpine limits the ability to draw in-depth conclusions about the drug’s direct, isolated effects on neuroimmune cell populations in the absence of traumatic brain injury. Despite the potential promising effects, the question of using reserpine in clinical practice as an anti-inflammatory drug, given its side effects, can only be resolved after large-scale clinical trials. Therefore, further investigations with longer follow-up durations in larger mouse cohorts, as well as randomized controlled trials, are warranted.

## 5. Conclusions

In this study, we have demonstrated the crucial role of bone marrow hematopoiesis and the mobilization of inflammatory cells from the bone marrow into the blood in the development of neuroinflammation in male ICR mice with TBI. Furthermore, granulocytic hematopoietic islands and hematopoietic precursors, along with the mobilization process, represent potential targets for modulating neuroinflammation. The therapeutic efficacy of the sympatholytic agent reserpine in ameliorating neuroinflammation following a TBI has been demonstrated. The mechanism of reserpine’s action is based on changes in hematopoiesis, the mobilization of inflammatory cells, and their migration into the brain, as well as the inhibition of microglial cells. Targeting the sympathetic regulation of hematopoiesis and cell mobilization can be considered a potential approach for mitigating neuroinflammation and degeneration in patients with TBI.

## Figures and Tables

**Figure 1 biomedicines-13-03080-f001:**
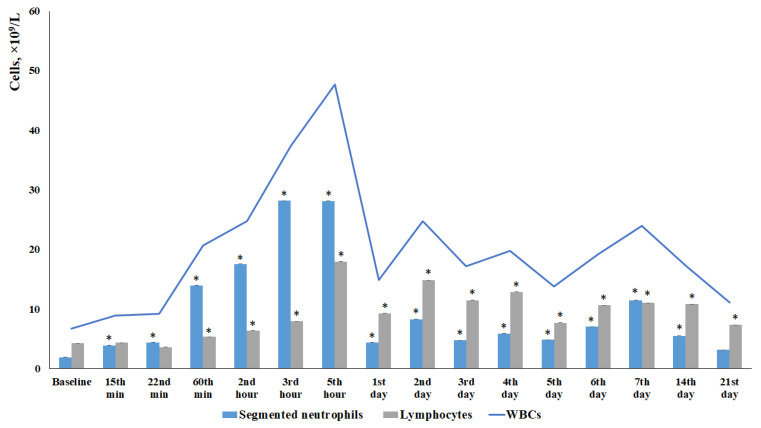
The effect of TBI on the hematological parameters of ICR mice: WBCs, segmented neutrophils and lymphocytes (×10^9^/L). * Significance of difference compared with baseline (*p* < 0.05).

**Figure 2 biomedicines-13-03080-f002:**
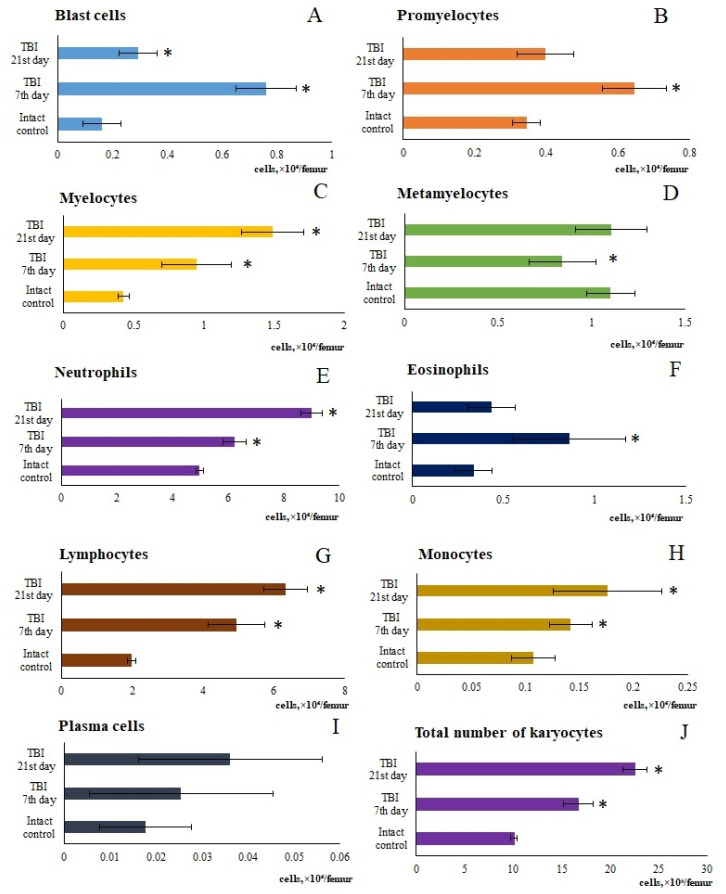
Changes in the content of individual morphologically recognizable forms of karyocytes in the bone marrow of male ICR mice following a TBI. The content of (**A**) blast cells, (**B**) promyelocytes, (**C**) myelocytes, (**D**) metamyelocytes, (**E**) neutrophils, (**F**) eosinophils, (**G**) lymphocytes, (**H**) monocytes, (**I**) plasma cells, and (**J**) total number of karyocytes. * Significance of difference compared with intact control (*p* < 0.05).

**Figure 3 biomedicines-13-03080-f003:**
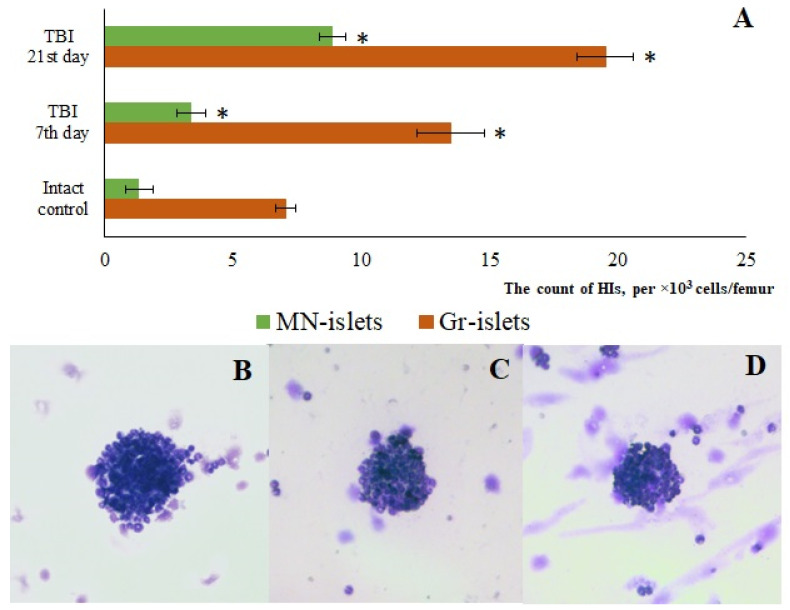
The effect of TBI on the number of macrophage-negative (MN) and granulocytic (Gr) islets in the bone marrow obtained from the femur of intact and TBI mice (7 and 21 days post-TBI). (**A**) The effect of TBI on the number of hematopoietic islets (per 10^3^/femur).* Significance of difference compared with intact control (*p* < 0.05). (**B**–**D**) Micrographs of bone marrow hematopoietic islets isolated from the femur of (**B**) intact mice, (**C**) TBI mice (21 day), and (**D**) TBI mice treated with reserpine (21 day). Azure and eosin staining, ×100.

**Figure 4 biomedicines-13-03080-f004:**
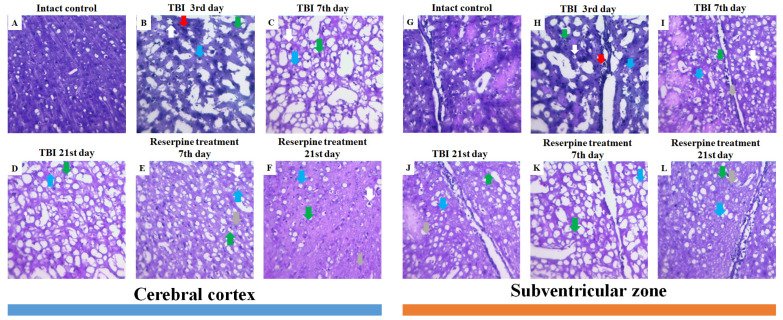
Representative micrographs of (**A**–**F**) cerebral cortex and (**G**–**L**) SVZ obtained from male ICR mice on 3rd, 7th, and 21st day after TBI. (**A**,**G**) Mice of intact control; (**B**,**H**) mice with TBI, 3rd day after injury; (**C**,**I**) with TBI, 7th day after injury; (**D**,**J**) with TBI, 21st day after injury; (**E**,**K**) mice with TBI treated with reserpine 7th day after injury; (**F**,**L**) mice with TBI treated with reserpine 21st day after injury. Tissues stained by hematoxylin and eosin, ×100. Red arrow—inflammatory infiltration; Green arrow—vacuolized glial cell; Blue arrow—vacuolized neuron; Grey arrow—glial mitosis; White arrow—the pycnotized neuron.

**Figure 5 biomedicines-13-03080-f005:**
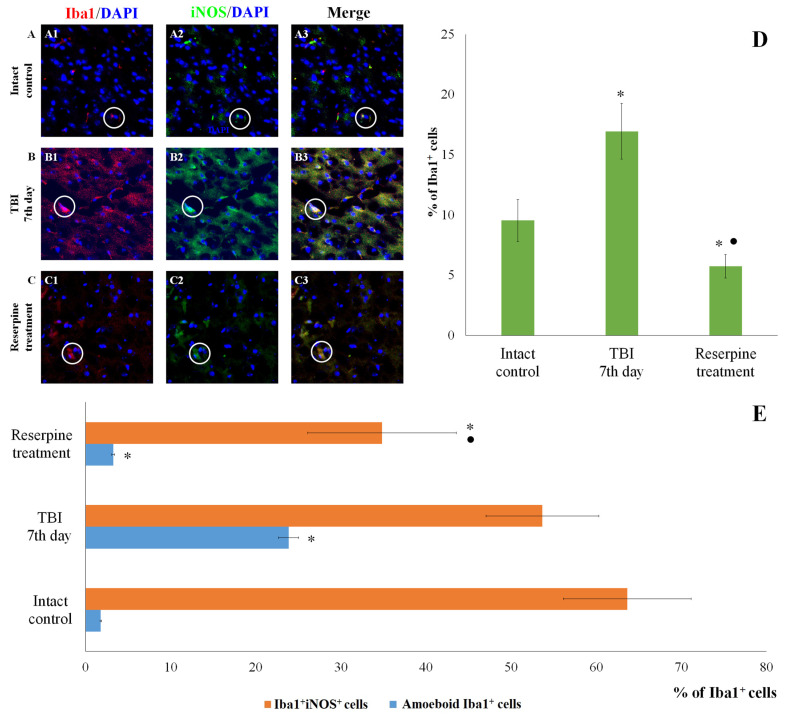
The count of Iba1^+^iNOS^+^ cells in the cerebral cortex of male ICR mice at day 7 post-TBI. (**A**–**C**) Representative micrographs of the cerebral cortex obtained from (**A1**–**A3**) intact mice; (**B1**–**B3**) TBI mice (day 7); and (**C1**–**C3**) TBI mice treated with reserpine. Samples were stained with antibodies specific for (**A1**,**B1**,**C1**) Iba1 (red), (**A2**,**B2**,**C2**) iNOS (green). DAPI (blue) was used to identify cell nuclei. (Merge) Composite image using all three colors. 4× images. (**D**) Number of Iba1^+^ cells (% of total DAPI-stained cells); (**E**) number of Iba1^+^iNOS^+^ cells and amoeboid Iba1^+^ cells (% of Iba1^+^ cells) in the cerebral cortex of male ICR mice. * Significance of difference compared with intact control (*p* < 0.017 with the Bonferroni correction). • Significance of difference compared with TBI (*p* < 0.017 with the Bonferroni correction). Note: The white circle represents the area of staining for the Iba1 and iNOS markers.

**Figure 6 biomedicines-13-03080-f006:**
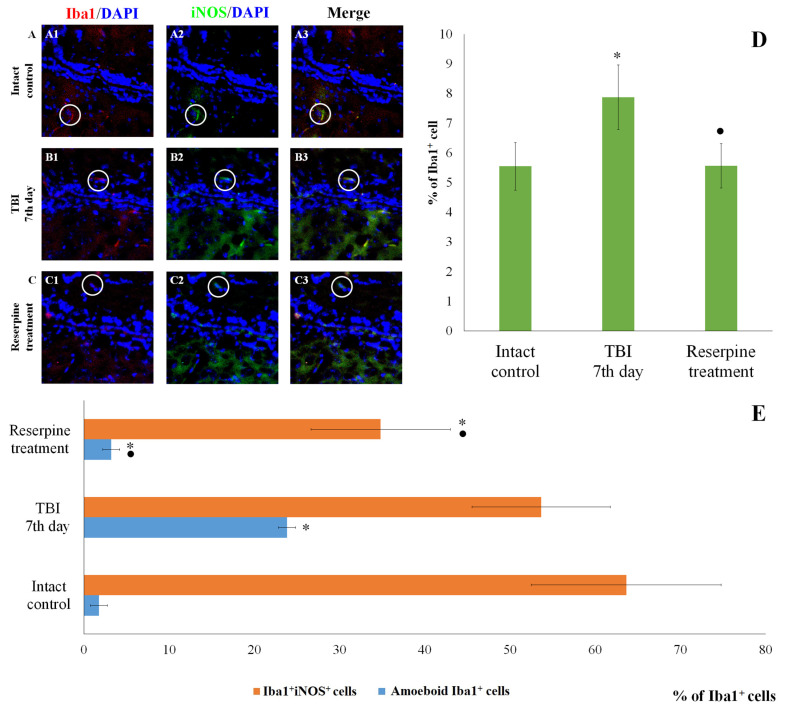
The count of Iba1^+^iNOS^+^ cells in the subventricular zone of male ICR mice at day 7 post-TBI. (**A**–**C**) Representative micrographs of the subventricular zone obtained from (**A1**–**A3**) intact mice; (**B1**–**B3**) TBI mice (day 7); and (**C1**–**C3**) TBI mice treated with reserpine. Samples were stained with antibodies specific for (**A1**,**B1**,**C1**) Iba1 (red); (**A2**,**B2**,**C2**) iNOS (green). DAPI (blue) was used to identify cell nuclei. (Merge) Composite image using all three colors. 4× images. (**D**) Number of Iba1^+^ cells (% of total cell number); (**E**) number of Iba1^+^iNOS^+^ cells and amoeboid Iba1^+^ cells (% of Iba1^+^ cells) in the cerebral cortex of male ICR mice. * Significance of difference compared with intact control (*p* < 0.017 with the Bonferroni correction). • Significance of difference compared with TBI (*p* < 0.017 with the Bonferroni correction). Note: The white circle represents the area of staining for the Iba1 and iNOS markers.

**Figure 7 biomedicines-13-03080-f007:**
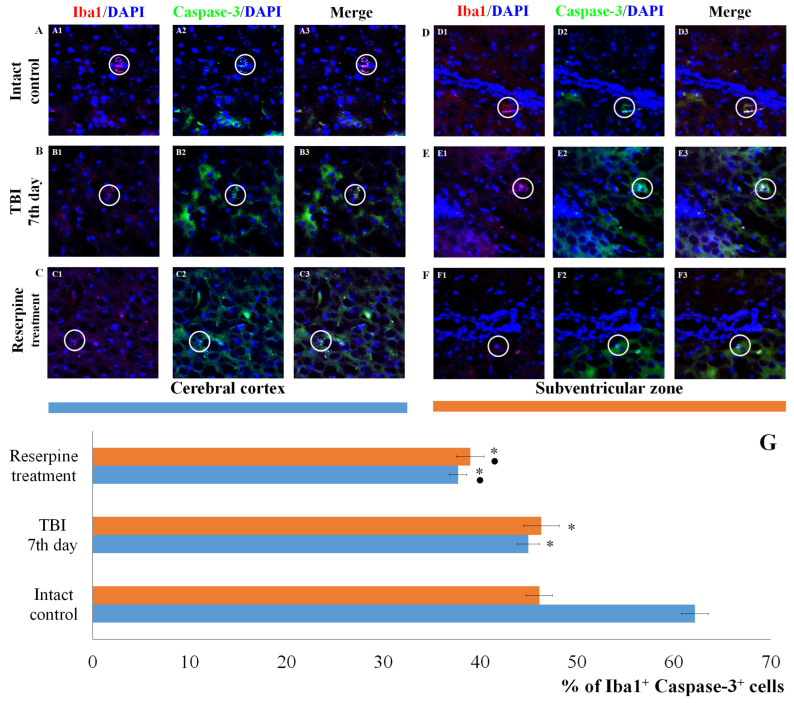
The count of Iba1^+^Caspase-3^+^ cells in the cerebral cortex and the subventricular zone of male ICR mice at day 7 post-TBI. Representative micrographs of the (**A**–**C**) cerebral cortex and (**D**–**F**) the subventricular zone obtained from (**A**,**D**) intact mice; (**B**–**E**) TBI mice (day 7); and (**C**–**F**) TBI mice treated with reserpine. Samples were stained with antibodies specific for (**A1**,**B1**,**C1**,**D1**,**E1**,**F1**) Iba1 (red); (**A2**,**B2**,**C2**,**D2**,**E2**,**F2**) Caspase-3 (green). DAPI (blue) was used to identify cell nuclei. (**A3**,**B3**,**C3**,**D3**,**E3**,**F3**) (Merge) Composite image using all three colors. 4× images. (**G**) Number of Iba1^+^Caspase-3^+^ cells (% of Iba1^+^ cells) in the cerebral cortex and subventricular zone of male ICR mice. * Significance of difference compared with intact control (*p* < 0.017 with the Bonferroni correction). • Significance of difference compared with TBI (*p* < 0.017 with the Bonferroni correction). Note: The white circle represents the area of staining for the Iba1 and Caspase-3 markers.

**Figure 8 biomedicines-13-03080-f008:**
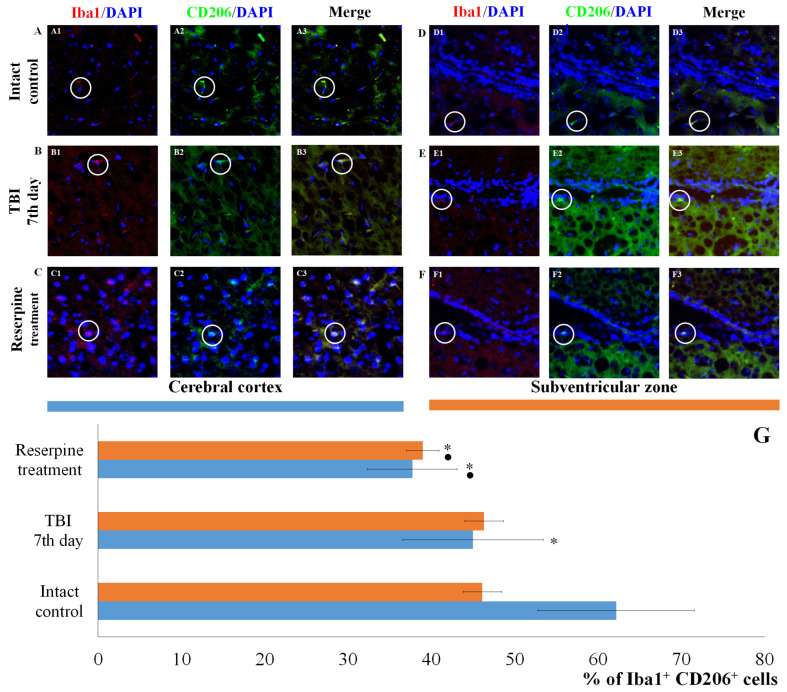
The count of Iba1^+^CD206^+^ cells in the cerebral cortex and subventricular zone of male ICR mice at day 7 post-TBI. Representative micrographs of the (**A**–**C**) cerebral cortex and (**D**–**F**) subventricular zone obtained from (**A**,**D**) intact mice; (**B**–**E**) TBI mice (day 7); and (**C**–**F**) TBI mice treated with reserpine. Samples were stained with antibodies specific to (**A1**,**B1**,**C1**,**D1**,**E1**,**F1**) Iba1 (red); (**A2**,**B2**,**C2**,**D2**,**E2**,**F2**) CD206^+^ (green). DAPI (blue) was used to identify cell nuclei. (**A3**,**B3**,**C3**,**D3**,**E3**,**F3**) (Merge) Composite image using all three colors. 4× images. (**G**) Number of Iba1^+^CD206^+^ cells (% of Iba1^+^ cells) in the cerebral cortex and subventricular zone of male ICR mice. * Significance of difference compared with intact control (*p* < 0.017 with the Bonferroni correction). • Significance of difference compared with TBI (*p* < 0.017 with the Bonferroni correction). Note: The white circle represents the area of staining for the Iba1 and CD206 markers.

**Figure 9 biomedicines-13-03080-f009:**
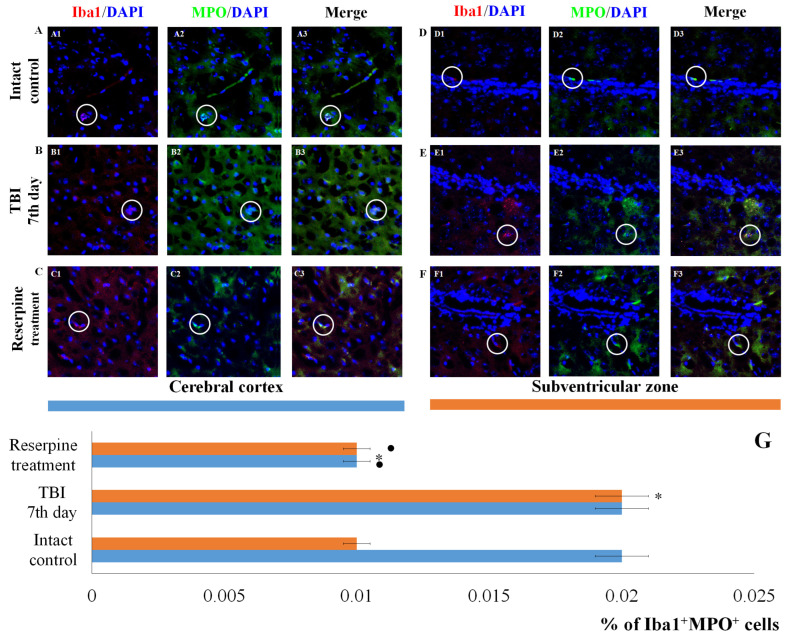
The count of Iba1^+^MPO^+^ cells in the cerebral cortex and subventricular zone of male ICR mice at day 7 post-TBI. Representative micrographs of the (**A**–**C**) cerebral cortex and (**D**–**F**) subventricular zone obtained from (**A**,**D**) intact mice; (**B**–**E**) TBI mice (day 7); and (**C**–**F**) TBI mice treated with reserpine. Samples were stained with antibodies specific to (**A1**,**B1**,**C1**,**D1**,**E1**,**F1**) Iba1 (red); (**A2**,**B2**,**C2**,**D2**,**E2**,**F2**) CD206^+^ (green). DAPI (blue) was used to identify cell nuclei. (**A3,B3,C3,D3,E3,F3**) (Merge) Composite image using all three colors. 4× images. (**G**) Number of Iba1^+^MPO^+^ cells (% of Iba1^+^ cells) in the cerebral cortex and subventricular zone of male ICR mice. * Significance of difference compared with intact control (*p* < 0.017 with the Bonferroni correction). • Significance of difference compared with TBI (*p* < 0.017 with the Bonferroni correction). Note: The white circle represents the area of staining for the Iba1 and MPO markers.

**Figure 10 biomedicines-13-03080-f010:**
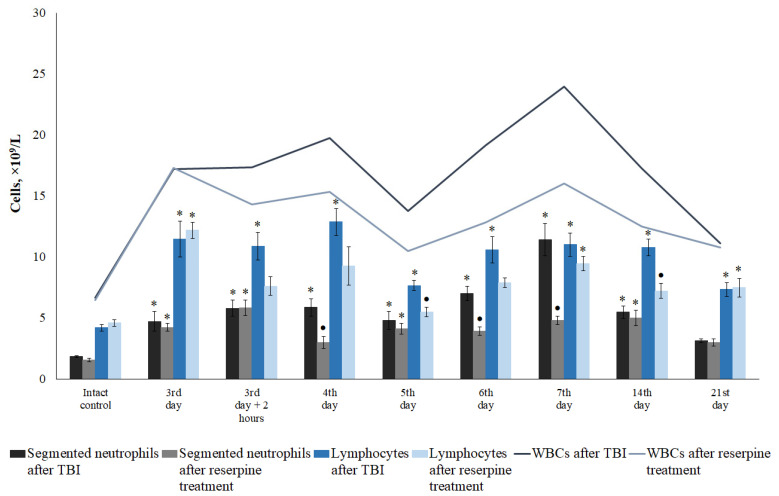
The effect of reserpine on the content of leukocytes (WBC), segmented neutrophils, and lymphocytes (10^9^/L) in the peripheral blood of TBI mice treated and untreated with reserpine. * Significance of difference compared with intact control (*p* < 0.017 with the Bonferroni correction). • Significance of difference compared with TBI (*p* < 0.017 with the Bonferroni correction).

**Figure 11 biomedicines-13-03080-f011:**
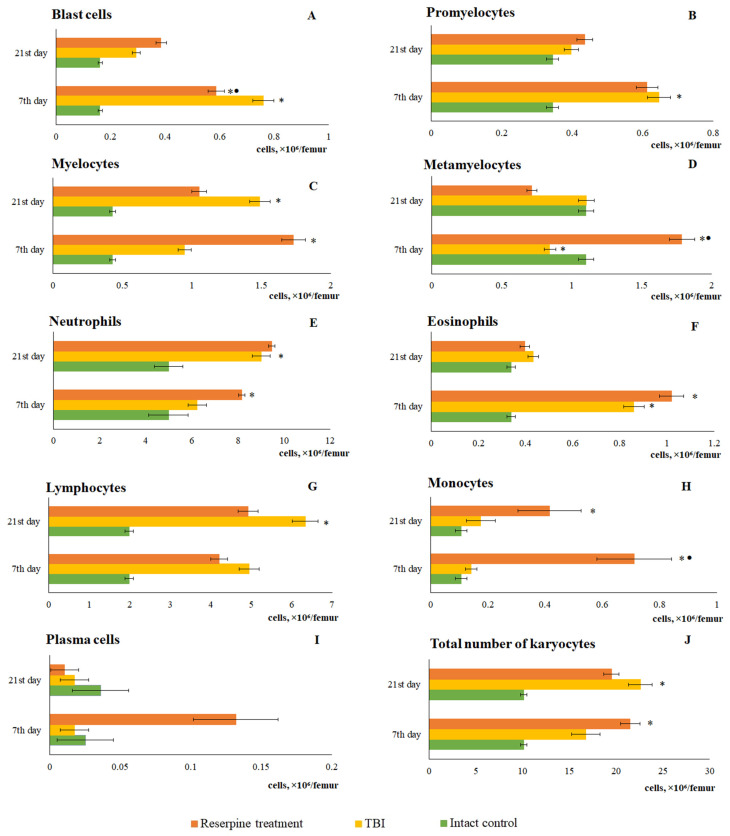
The effect of reserpine on the content of individual morphologically recognizable karyocyte forms in the bone marrow of ICR mice after TBI. The content of (**A**) blast cells, (**B**) promyelocytes, (**C**) myelocytes, (**D**) metamyelocytes, (**E**) neutrophils, (**F**) eosinophils, (**G**) lymphocytes, (**H**) monocytes, (**I**) plasma cells, (**J**) total number of karyocytes. * Significance of difference compared with intact control (*p* < 0.017 with the Bonferroni correction). • Significance of difference compared with TBI (*p* < 0.017 with the Bonferroni correction).

**Figure 12 biomedicines-13-03080-f012:**
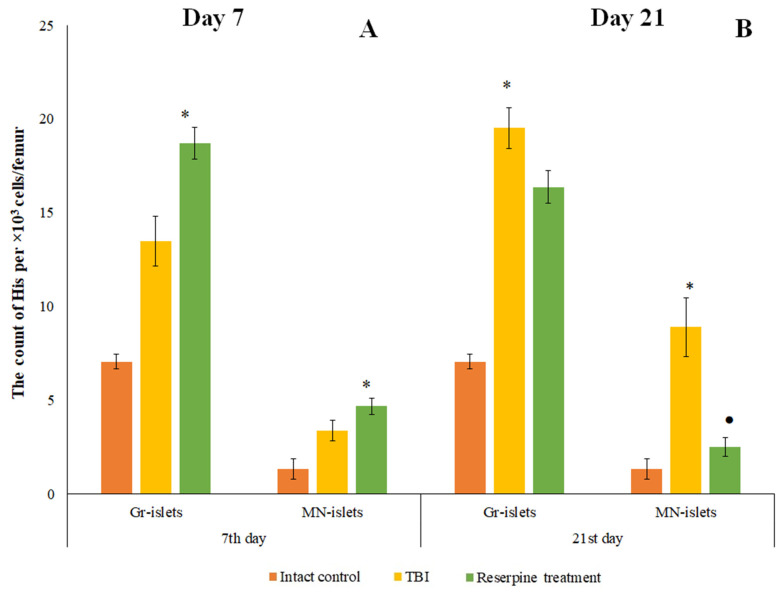
The effect of reserpine treatment on the granulocytic (Gr) and macrophage-negative (MN) islets in the bone marrow obtained from the femur of intact mice and TBI mice on (**A**) 7 and (**B**) 21 days post-TBI). * Significance of difference compared with intact control (*p* < 0.017 with the Bonferroni correction). • Significance of difference compared with TBI (*p* < 0.017 with the Bonferroni correction).

**Figure 13 biomedicines-13-03080-f013:**
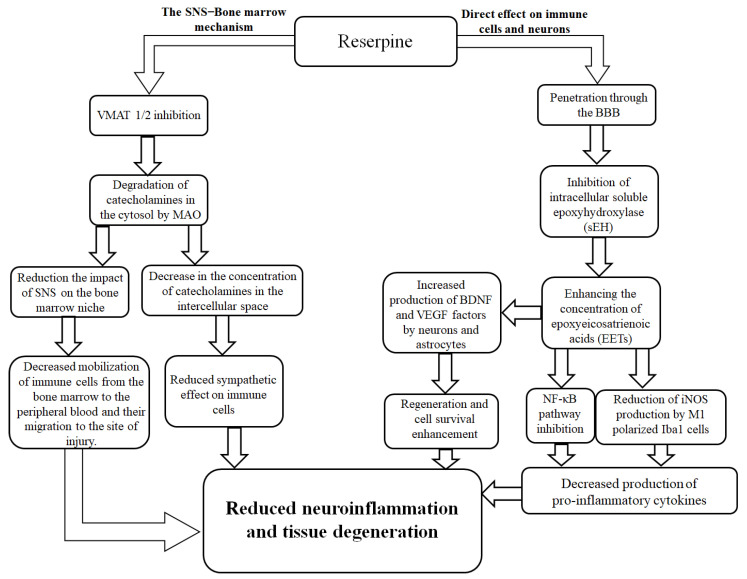
Possible mechanisms of direct and indirect effects of the sympatholytic drug reserpine on neuroinflammation after TBI. Reserpine exerts its effects via two potential mechanisms. The first mechanism is mediated through the attenuation of the sympathetic nervous system’s (SNS) influence on the bone marrow and immune cells. This occurs via the inhibition of vesicular monoamine transporters 1 and 2 (VMAT 1/2), leading to subsequent degradation of catecholamines in the cytosol by monoamine oxidase (MAO). The resultant decrease in catecholamine concentration in the intercellular space reduces the sympathetic tone on the bone marrow niche and immune cells. This ultimately diminishes the mobilization of immune cells from the bone marrow to the peripheral blood and their migration to the injury site. The second mechanism involves the direct action of reserpine on microglial cells in the brain. Upon penetrating the blood–brain barrier (BBB), reserpine inhibits intracellular soluble epoxide hydrolase (sEH). This inhibition enhances the concentration of epoxyeicosatrienoic acids (EETs), which subsequently suppress the pro-inflammatory NF-κB signaling pathway. This leads to a reduction in inducible nitric oxide synthase (iNOS) production by M1-polarized microglia (Iba1+ cells), thereby decreasing the production of pro-inflammatory cytokines. Concurrently, EETs promote the production of brain-derived neurotrophic factors (BDNFs) and vascular endothelial growth factors (VEGFs) by neurons and astrocytes, enhancing cell survival and regeneration. Collectively, these actions result in reduced neuroinflammation and tissue degeneration.

**Table 1 biomedicines-13-03080-t001:** The types of samples and sample collection points.

Sample	Investigation	Points of Sample Collection
blood	The white blood cells count Leukocyte formula	−1 d, 15 min, 22 min, 60 min, 2 h, 3 h, 5 h, 1 d, 2 d, 3 d, 4 d, 5 d 6 d, 7 d, 14 d, 21 d
bone marrow	Total number of karyocytes	5 h, 1 d, 3 d, 7 d, 21 d
	Myelography	7 d, 21 d
	Hematopoietic islands	7 d, 21 d
	In vitro studies	1 h, 5 h, 1 d, 3 d, 7 d, 14 d, 21 d
brain	Histological Examination	3 d, 7 d, 21 d
	Immunohistochemical Examination	7 d

**Table 2 biomedicines-13-03080-t002:** The dynamics of band neutrophil, eosinophil, monocytes, plasma cell counts (×10^9^/L) in the peripheral blood of male ICR mice after TBI (M ± m).

Time	Band Neutrophils	Eosinophils	Monocytes	Plasma Cells
Baseline	0.24 ± 0.05	1.87 ± 0.07	0.00 ± 0.00	0.00 ± 0.00
15th min	**0.14 ± 0.03 ***	**3.88 ± 0.17 ***	0.00 ± 0.00	**0.11 ± 0.05 ***
22nd min	**0.14 ± 0.04 ***	**4.36 ± 0.63 ***	0.00 ± 0.00	**0.05 ± 0.02 ***
60th min	**0.10 ± 0.08***	**13.97 ± 0.80 ***	0.00 ± 0.00	**0.16 ± 0.05 ***
2nd hour	**0.04 ± 0.04 ***	**17.53 ± 1.05 ***	0.00 ± 0.00	**0.09 ± 0.05 ***
3rd hour	0.28 ± 0.11	**28.15 ± 1.80 ***	0.00 ± 0.00	**0.22 ± 0.09 ***
5th hour	0.31 ± 0.13	**28.13 ± 2.43 ***	0.00 ± 0.00	**0.12 ± 0.08 ***
1st day	**0.56 ± 0.08 ***	**4.40 ± 0.41 ***	**0.04 ± 0.02 ***	**0.15 ± 0.06 ***
2nd day	**0.63 ± 0.11 ***	**8.28 ± 0.89 ***	0.00 ± 0.00	**0.21 ± 0.13 ***
3rd day	0.29 ± 0.10	**4.73 ± 0.81 ***	0.00 ± 0.00	0.00 ± 0.00
4th day	0.34 ± 0.08	**5.89 ± 0.71 ***	**0.04 ± 0.04 ***	**0.13 ± 0.06 ***
5th day	0.35 ± 0.13	**4.82 ± 0.73 ***	0.00 ± 0.00	**0.04 ± 0.02 ***
6th day	**0.50 ± 0.11 ***	**7.01 ± 0.58 ***	0.00 ± 0.00	**0.11 ± 0.08 ***
7th day	**0.62 ± 0.06 ***	**11.45 ± 1.33 ***	0.00 ± 0.00	0.00 ± 0.00
14th day	0.36 ± 0.07	**5.48 ± 0.50 ***	0.00 ± 0.00	**0.23 ± 0.12 ***
21st day	0.19 ± 0.07	**3.14 ± 0.16 ***	0.00 ± 0.00	**0.22 ± 0.05 ***

* Significance of difference compared with baseline (*p* < 0.05).

**Table 3 biomedicines-13-03080-t003:** Colony-forming unit granulocyte (CFU-G) activity in bone marrow of TBI mice treated and untreated with reserpine (colonies per ×10^5^ cells/femur) (M ± m).

Time	TBI	Reserpine
Intact control	2.67 ± 0.13	
60th min	**8.33 ± 0.42 ***	-
5th hour	**6.17 ± 0.31 ***	-
1st day	**10.83 ± 0.54 ***	-
3rd day	**32.50 ± 1.63 ***	**33.83 ± 1.69 ***
7th day	2.50 ± 0.13	3.67 ± 0.18
14th day	2.33 ± 0.12	1.00 ± 0.05
21st day	3.00 ± 0.15	4.67 ± 0.23

* Significance of difference compared with intact control (*p* < 0.017 with the Bonferroni correction).

**Table 4 biomedicines-13-03080-t004:** The number of band neutrophils, eosinophils, monocytes, plasma cells (10^9^/L) in the peripheral blood of TBI mice treated and untreated with reserpine (M ± m).

Time	Band Neutrophils		Eosinophils		Monocytes		Plasma Cells	
	TBI	Reserpine	TBI	Reserpine	TBI	Reserpine	TBI	Reserpine
Intact control	0.24 ± 0.05	0.12 ± 0.03	0.10 ± 0.03	0.17 ± 0.05	0.00 ± 0.00	0.00 ± 0.00	0.00 ± 0.00	0.00 ± 0.00
3rd day	**0.29 ± 0.10 ***	**0.33 ± 0.11 ***	**0.70 ± 0.24 ***	**0.31 ± 0.07 *●**	0.00 ± 0.00	0.00 ± 0.00	0.00 ± 0.00	**0.00 ± 0.00**
3rd day + 2 h	0.22 ± 0.08	**0.25 ± 0.06 ***	**0.41 ± 0.10 ***	**0.43 ± 0.13 ***	0.00 ± 0.00	0.00 ± 0.00	0.00 ± 0.00	**0.12 ± 0.03 *●**
4th day	**0.34 ± 0.08 ***	**0.28 ± 0.09 *●**	**0.50 ± 0.18 ***	**0.63 ± 0.25 *●**	**0.04 ± 0.04 ***	**0.01 ± 0.01 ***	**0.13 ± 0.06 ***	**0.12 ± 0.06 ***
5th day	**0.35 ± 0.13 ***	**0.30 ± 0.08 *●**	**0.91 ± 0.22 ***	**0.47 ± 0.11 *●**	0.00 ± 0.00	**0.02 ± 0.02 ***	**0.04 ± 0.02 ***	**0.08 ± 0.04 *●**
6th day	**0.50 ± 0.11 ***	**0.32 ± 0.08 *●**	**0.88 ± 0.24 ***	**0.55 ± 0.16 *●**	0.00 ± 0.00	0.00 ± 0.00	**0.11 ± 0.08 ***	**0.13 ± 0.04 *●**
7th day	**0.62 ± 0.06 ***	**0.50 ± 0.10 *●**	**0.91 ± 0.45 ***	**1.02 ± 0.26 *●**	0.00 ± 0.00	0.00 ± 0.00	0.00 ± 0.00	**0.20 ± 0.06 *●**
14th day	**0.36 ± 0.07 ***	**0.23 ± 0.08 *●**	**0.42 ± 0.18 ***	**0.38 ± 0.08 ***	0.00 ± 0.00	0.00 ± 0.00	**0.23 ± 0.12 ***	**0.24 ± 0.08 ***
21st day	0.19 ± 0.07	**0.10 ± 0.06●**	**0.18 ± 0.07 ***	**0.07 ± 0.04 *●**	0.00 ± 0.00	0.00 ± 0.00	**0.22 ± 0.0 *5**	**0.10 ± 0.06 *●**

* Significance of difference compared with intact control (*p* < 0.017 with the Bonferroni correction). • Significance of difference compared with TBI (*p* < 0.017 with the Bonferroni correction).

## Data Availability

The data presented in this publication can be accessed by contacting the corresponding author, subject to a reasonable request.

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
