# Peer review of "Sympathetic Regulation of Hematopoiesis and the Mobilization of Inflammatory Cells in ICR Mice with Traumatic Brain Injury: A Novel Approach to Targeting Neuroinflammation and Degenerative Processes"

_biomedicines, 2025, doi:10.3390/biomedicines13123080_

Round 1
Reviewer 1 Report
Comments and Suggestions for Authors
Dear Authors,
This is an innovative preclinical study investigating the sympathetic nervous system's (SNS) role in propagating neuroinflammation post-traumatic brain injury (TBI) via regulation of hematopoietic processes. The authors successfully demonstrate that TBI activates hematopoiesis and inflammatory cell mobilization , and that treatment with the sympatholytic drug reserpine effectively reduces leukocytosis, hematopoietic activity, and key markers of neuroinflammation (M1-polarized microglia, Caspase-3+, MPO-expressing cells) and degenerative changes in the cerebral cortex and subventricular zone. The work provides a compelling novel mechanism for TBI treatment.
- Reserpine Dosing and Timing Rationale: The study uses a variable, low-dose regimen for reserpine administration (an initial dose of 1.0 mg/kg followed by 0.1 mg/kg on days 3–7). This specific dosage and the timing of the intervention (starting on Day 3 post-injury) are critical to the study's conclusions. The authors should strengthen the justification in the Methods and Discussion sections for why this specific regimen was selected, particularly relating the initial higher dose and the delayed start to the known pharmacokinetics of reserpine and the timeline of TBI pathology in mice.
- Discussion: improve discussion
M2 Microglia Interpretation: The abstract states that reserpine decreased the number of M2-polarized microglial cells in the SVZ. M2 polarization is typically associated with the anti-inflammatory, tissue-repair phase. The Discussion should explicitly address and interpret this finding. A reduction in the "repair" phenotype (M2) while concluding an overall positive anti-neuroinflammatory effect is a complex result that requires specific commentary to ensure the net effect of reserpine is fully understood.
Comments on the Quality of English Languageminor grammar editing
Author Response
Comments 1: Reserpine Dosing and Timing Rationale: The study uses a variable, low-dose regimen for reserpine administration (an initial dose of 1.0 mg/kg followed by 0.1 mg/kg on days 3–7). This specific dosage and the timing of the intervention (starting on Day 3 post-injury) are critical to the study's conclusions. The authors should strengthen the justification in the Methods and Discussion sections for why this specific regimen was selected, particularly relating the initial higher dose and the delayed start to the known pharmacokinetics of reserpine and the timeline of TBI pathology in mice.
Response 1: Thank you for the question. Reserpine doses were selected based on our previous studies demonstrating reserpine activity, including anti-inflammatory, in various pathologies (diabetes mellitus, idiopathic pulmonary fibrosis, myelosuppression) at similar doses [1-4], as well as studies related to the effects of reserpine in various doses on the central nervous system [5,6]
The choice of reserpine administration regimen in this study was based on the results of preliminary research, which demonstrated the development of the most significant changes in the blood (leukocytosis) and brain (microglial activation, neuronal death, and neuroinflammation) on day 3 post-TBI.
We have added this information to the "Materials and Methods", lines 136-139 and "Discussion" section, lines 572-586.
References:
- Skurikhin, E.G.; Pershina, O.V.; Ermakova, N.N.; Ermolaeva, L.A.; Krupin, V.A.; et al. Polytherapy with Reserpine and Glucagon-like Peptide-1 (GLP1) Improves the Symptoms in Streptozotocin-Induced Type-1 Diabetic Mice by Reducing Inflammation and Inducting Beta Cell Regeneration. Stem Cell Res. Ther. 2018. 8, 434. doi: 10.4172/2157-7633.1000434
- Skurikhin, E.G.; Khmelevskaya, E.S.; Pershina, O.V.; Ermakova, N.N.; Krupin, V.A.; et al. Anti-Fibrotic Effects of Reserpine on Lung Fibrosis: Stem Cells in the Pathogenesis of Pneumofibrosis. Open Conf. Proc. J. 2013. 7, 1, doi:10.2174/2210289201304010031.
- Skurikhin, E.G.; Ermakova, N.N.; Pershina, O.V.; Krupin, V.A.; Pakhomova, A.V.; Dygai, A.M. Response of Hematopoietic Stem and Progenitor Cells to Reserpine in C57Bl/6 Mice. Bull. Exp. Biol. Med. 2016, 160, 439–443, doi:10.1007/s10517-016-3191-y.
- Skurikhin, E.G.; Pershina, O.V.; Reztsova, A.M.; Ermakova, N.N.; Khmelevskaya, E.S.; et al. Modulation of bleomycin-induced lung fibrosis by pegylated hyaluronidase and dopamine receptor antagonist in mice. PLoS One. 2015, 10, e0125065. doi: 10.1371/journal.pone.0125065.
- Strawbridge, R.; Javed, R.R.; Cave, J.; Jauhar, S.; Young, A.H. The Effects of Reserpine on Depression: A Systematic Re-view. J. Psychopharmacol. (Oxf.) 2022, doi:10.1177/02698811221115762.
- Weir, M.R. Reserpine: A New Consideration of an Old Drug for Refractory Hypertension. Am. J. Hypertens. 2020, 33, 708–710, doi:10.1093/ajh/hpaa069.
Comments 2: M2 Microglia Interpretation: The abstract states that reserpine decreased the number of M2-polarized microglial cells in the SVZ. M2 polarization is typically associated with the anti-inflammatory, tissue-repair phase. The Discussion should explicitly address and interpret this finding. A reduction in the "repair" phenotype (M2) while concluding an overall positive anti-neuroinflammatory effect is a complex result that requires specific commentary to ensure the net effect of reserpine is fully understood.
Response 2: Thank you for your comment. Indeed, reserpine decreased the number of M2 polarized microglia in the SVZ without affecting their content in the MC. This ambiguous effect of reserpine may be related to the contradictory influence of catecholamines on immune cells. It is believed that the impact of stimulation of immune cell adrenergic receptors depends on such factors as the initial activity of the cells [1], the proximity of the cell to the source of catecholamines, cytokines and chemokines [2], and the density of adrenergic receptor expression on its surface [3]. Another potential mechanism of the anti-inflammatory effect of reserpine is its inhibition of soluble epoxyhydroxylase (sEH) [4]. This inhibition reduces the activation of the pro-inflammatory NF-κB pathway, leading to a decrease in the production of pro-inflammatory cytokines such as TNF-α [5]. It also reduces the expression of inducible nitric oxide synthase (iNOS), which is characteristic of M1-polarized microglia [6].
We suggest that a reduction in the number of M2 microglia may be associated with a modulation of inflammation in response to reserpine treatment. This modulation could occur either through the direct migration of anti-inflammatory cells to the damaged area or indirectly through changes in the concentration of catecholamines in the interstitial space, which affect the catecholaminergic tone of the sympathetic nervous system and the brain.
According to the reviewer's request, we have expanded our discussion of the effects of reserpine on M2-polarized microglial cells in "Discussion" section, lines 572-588.
References:
- Straub, R.H.; Rauch, L.; Fassold, A.; Lowin, T.; Pongratz, G. Neuronally Released Sympathetic Neurotransmitters Stimulate Splenic Interferon-γ Secretion from T Cells in Early Type II Collagen–Induced Arthritis. Arthritis Rheum. 2008, 58, 3450–3460, doi:10.1002/art.24030.
- Li, W.; Knowlton, D.; Woodward, W.R.; Habecker, B.A. Regulation of Noradrenergic Function by Inflammatory Cytokines and Depolarization. J. Neurochem. 2003, 86, 774–783.
- Lorton, D.; Lubahn, C.; Bellinger, D.L. Potential Use of Drugs That Target Neural-Immune Pathways in the Treatment of Rheumatoid Arthritis and Other Autoimmune Diseases. Curr. Drug Targets Inflamm. Allergy 2003, 2, 1–30.
- Verma, K.; Paliwal, S.; Sharma, S. Therapeutic Potential of Reserpine in Metabolic Syndrome: An Evidence Based Study. Pharmacol. Res. 2022, 186, 106531, doi:10.1016/j.phrs.2022.106531.
- Pallàs, M.; Vázquez, S.; Sanfeliu, C.; Galdeano, C.; Griñán-Ferré, C. Soluble Epoxide Hydrolase Inhibition to Face Neu-roinflammation in Parkinson’s Disease: A New Therapeutic Strategy. Biomolecules 2020, 10, 703, doi:10.3390/biom10050703.
- Xiao, H.; Yang, K. Role of Macrophages in Neuroimmune Regulation. Front. Immunol. 2025, 16, 1573174, doi:10.3389/fimmu.2025.1573174.
Reviewer 2 Report
Comments and Suggestions for Authors
Thank you for submitting this interesting work exploring a novel therapeutic approach for TBI-induced neuroinflammation. Your hypothesis that targeting sympathetic regulation of bone marrow hematopoiesis can modulate neuroinflammation is innovative and addresses an important gap in TBI treatment. The experimental design spanning blood, bone marrow, and brain analysis across multiple time points is commendable.
The manuscript demonstrates several strengths such as (1) Novel therapeutic target addressing an unmet clinical need. (2) Comprehensive temporal analysis (15 minutes to 21 days). (3) Multi-tissue approach (blood, bone marrow, brain regions). (4) Use of both histological and immunohistochemical methods and (5) Clear presentation of complex hematological data
However, to strengthen your manuscript for publication, several areas require attention
Major comments
1. Mechanistic clarity and experimental design
While you propose that reserpine works through sympathetic modulation of bone marrow hematopoiesis, the direct mechanistic link remains somewhat unclear.
Consider:
a) The reserpine treatment begins on day 3 post-TBI, yet you observe effects as early as 2 hours after the first injection. This rapid response raises questions about whether the effects are solely mediated through bone marrow modulation or if reserpine has more direct effects on existing circulating cells or brain-resident cells. Could you discuss this temporal paradox more explicitly?
b) Have you considered including a group receiving reserpine prior to TBI or immediately after injury? This would help clarify whether the drug’s effects are primarily preventive or therapeutic and would strengthen the mechanistic argument.
2. Statistical analysis and sample sizes
Several concerns arise regarding the statistical approach:
a) While you mention using 140 mice total, the exact sample sizes for each experimental group at each time point are not clearly specified. Given the large number of time points (15 minutes to 21 days) and multiple analyses, please clarify:
- How many animals were analyzed per group per time point?
- Were power calculations performed to determine sample sizes?
- How were animals distributed across the different analyses (blood, bone marrow, histology)?
b) The statement “A p-value of less than 0.05 (by two-tailed testing) was considered an indicator of statistical significance” is noted, but there’s no mention of correction for multiple comparisons. With numerous time points and outcome measures, this is a concern. Have you considered applying Bonferroni or false discovery rate corrections?
c) For the immunohistochemical analyses, you mention “5 brain sections” and “3 fields of view”. Is this n=1 animal with technical replicates, or were multiple animals examined? This distinction is crucial for statistical validity.
3. Experimental controls and groups
a) A reserpine-only group (without TBI) is missing. This control is essential to distinguish between reserpine’s effects on normal hematopoiesis versus TBI-induced changes. Does reserpine alter baseline bone marrow function or brain inflammation markers in healthy animals?
b) A vehicle control group is mentioned for TBI mice, but it is unclear if intact mice also received vehicle injections. This would control for the stress of repeated injections.
c) Have you considered using an alternative sympatholytic agent or beta-blocker as a comparison group? This would strengthen the argument that effects are specifically due to sympathetic modulation rather than off-target effects of reserpine
4. Blood-brain barrier considerations
An important question that deserves more discussion: To what extent can reserpine cross the blood-brain barrier, and what are its direct effects on brain-resident cells?
a) Reserpine can cross the BBB and directly affect central catecholamine systems. How do you distinguish between peripheral effects (bone marrow modulation) and direct CNS effects on microglia and neuroinflammation?
b) The reduction in Iba1+ cells and microglial activation could partially result from direct reserpine effects on brain-resident immune cells rather than solely through reduced peripheral cell infiltration.
c) Could you include markers that distinguish brain-resident microglia from infiltrating peripheral macrophages (e.g., TMEM119, P2RY12), though this might be challenging.
Minor comments
Writing and Presentation:
- Line 86-87: “All manipulations with animals were maintainedin accordance with he European Convention” - “maintained in accordance with the European Convention”
- Line 288: “TBI redused the total number of cells” - should be “reduced”
- The abbreviations list at the end is helpful, but CFU-G appears in the text before being defined
- Figure quality: Some of the immunofluorescence images (particularly Figures 5-9) would benefit from higher magnification insets to show cellular morphology more clearly or show where the microglia are with arrows or arrow heads.
- The term “pathological control” is used to refer to TBI mice without treatment - consider using “TBI control” or “untreated TBI” for clarity
Methods:
- Immunohistochemistry: You mention using “recombinant mouse monoclonal antibodies” and “rabbit polyclonal antibodies” - please specify if these are commercially available and include catalog numbers consistently (some are included, some are not)
- Line 196-198: The quantification method for IHC states “3 fields of view (200 × 200 μm area each)”. were these fields randomly selected or selected based on specific criteria? This affects potential bias
- Statistical methods: You mention Student’s t-test for parametric data and Mann-Whitney for nonparametric, but you don’t specify how normality was assessed
Results:
- Table 2 and Table 4: The presentation would be clearer if you bolded or highlighted significant values for easier identification
- Figure 1: The graph shows data points from 15 min to 21 days, but the scale transition from hours to days is abrupt. Consider using a log scale or breaking the axis for better visualization
- Figure 2: Error bars appear very small for some measurements, please confirm these are SEM as stated and not SD. Could also justify why males?
- Line 316-317: You state “Expression of the apoptotic marker Caspase-3 increased in Iba1+ cells of the SVZ” but the quantification in Figure 7G shows this as a percentage of Iba1+ cells, not absolute numbers. Please clarify whether total Caspase-3+ cell numbers also increased
Discussion:
- Lines 433-445: The discussion of conflicting literature on TBI-induced immune changes is good, but could be strengthened by discussing potential reasons for discrepancies (species differences, injury models, severity, timing)
- Lines 492-511: The mechanism discussion would benefit from a clearer model or schematic figure showing your proposed pathway: SNS activation -> bone marrow hematopoiesis -> cell mobilization -> brain infiltration -> neuroinflammation
- The discussion would benefit from addressing the clinical feasibility of reserpine for TBI, given its known side effects (depression, sedation, hypotension) and current limited use in medicine
References:
- References 23 and 24 are cited in the Methods section (line 133) but appear much later in the reference list. Please ensure citations are numbered in order of appearance
- Some references appear to be preprints (e.g., reference 31: Soliman et al., BioRxiv). This published in peer-reviewed journal, please update.
Figures and Tables:
- Figure 4: The hematoxylin and eosin images would benefit from arrows or annotations indicating specific features mentioned in the text (inflammatory infiltrates, hemorrhages, vacuolization)
- Figures 5-9: Consider providing single-channel images in the supplementary materials for readers who may have difficulty interpreting merged fluorescence images
Author Response
Comments 1a: Mechanistic clarity and experimental design
While you propose that reserpine works through sympathetic modulation of bone marrow hematopoiesis, the direct mechanistic link remains somewhat unclear.
The reserpine treatment begins on day 3 post-TBI, yet you observe effects as early as 2 hours after the first injection. This rapid response raises questions about whether the effects are solely mediated through bone marrow modulation or if reserpine has more direct effects on existing circulating cells or brain-resident cells. Could you discuss this temporal paradox more explicitly?
Response 1a: Thank you for your attention to that detail. We believe that this paradoxical effect of reserpine is due to its biphasic action and influence on the sympathetic innervation of the bone marrow niche. The first phase of reserpine action (3-5 min after administration) is accompanied by an increase in the level of free monoamines [1,2]. This short-term increase in monoamine concentration is sufficient to indirectly stimulate cell mobilization from the bone marrow. The second phase of reserpine's action (~2-4 hours after administration) is characterized by the depletion of the labile reserve of monoamines and elimination of the sympathetic influence on cell mobilization [1,2]. However, an indirect effect of reserpine on circulating cells and brain cells associated with changes in catecholamine concentrations cannot be ruled out.
References:
- Wimalasena, K. Vesicular monoamine transporters: structure-function, pharmacology, and medicinal chemistry. Med Res Rev. 2011. 31. 483-519. doi: 10.1002/med.20187.
- Skurikhin, E.G.; Ermakova, N.N.; Pershina, O.V.; Krupin, V.A.; Pakhomova, A.V.; Dygai, A.M. Response of Hematopoietic Stem and Progenitor Cells to Reserpine in C57Bl/6 Mice. Bull Exp Biol Med. 2016, 160, 439-443. doi: 10.1007/s10517-016-3191-y.
Comments 1b: Have you considered including a group receiving reserpine prior to TBI or immediately after injury? This would help clarify whether the drug’s effects are primarily preventive or therapeutic and would strengthen the mechanistic argument.
Response 1b: Thank you for your valuable suggestion. We have already finished several series of experiments to evaluate the effectiveness of other reserpine administration regimens. The results are currently being processed.
Comments 2a: Statistical analysis and sample sizes
Several concerns arise regarding the statistical approach:
While you mention using 140 mice total, the exact sample sizes for each experimental group at each time point are not clearly specified. Given the large number of time points (15 minutes to 21 days) and multiple analyses, please clarify: How many animals were analyzed per group per time point? Were power calculations performed to determine sample sizes? How were animals distributed across the different analyses (blood, bone marrow, histology)?
Response 2a: Thank you for your question. The power calculations were performed in G*Power. However, the study used the number of animals approved by the local ethics committee.
Moreover, we clarified the number of animals and their distribution among groups. A total of 136 animals were divided into three experimental groups: 56 mice - the intact control group, 56 mice - the TBI group, and 24 animals - the TBI group treated with reserpine. One day before TBI, blood was collected from all animals for analysis to assess baseline levels. Hematological parameters were monitored according to the schedule until day 21 of the experiment. Beginning at 1 hour (for the intact control and TBI groups) or from day 3 (for TBI mice treated with reserpine) after TBI, animals were gradually sacrificed from the experiment for bone marrow and brain examination. Bone marrow and brain samples were collected according to the experimental schedule from 8 mice from each group. The detailed distribution of animals by groups and time points is as follows:
|
Time |
Sample |
Number of animals |
||
|
Intact control |
TBI |
Reserpine treatment |
||
|
-1 d |
Blood |
56 |
56 |
24 |
|
15 min |
56 |
56 |
24 |
|
|
22 min |
56 |
56 |
24 |
|
|
1 h |
Blood |
56 |
56 |
24 |
|
Bone marrow |
8 |
8 |
- |
|
|
2 h |
Blood |
48 |
48 |
24 |
|
5 h |
Blood |
48 |
48 |
24 |
|
Bone marrow |
8 |
8 |
- |
|
|
1 d |
Blood |
40 |
40 |
24 |
|
Bone marrow |
8 |
8 |
- |
|
|
2 d |
Blood |
40 |
40 |
24 |
|
3 d |
Blood |
40 |
40 |
24 |
|
Bone marrow |
8 |
8 |
- |
|
|
Brain |
8 |
8 |
- |
|
|
4d |
Blood |
32 |
32 |
24 |
|
5d |
Blood |
32 |
32 |
24 |
|
6d |
Blood |
32 |
32 |
24 |
|
7d |
Blood |
24 |
24 |
24 |
|
Bone marrow |
8 |
8 |
8 |
|
|
Brain |
8 |
8 |
8 |
|
|
14 d |
Blood |
16 |
16 |
16 |
|
Bone marrow |
8 |
8 |
8 |
|
|
Brain |
8 |
8 |
8 |
|
|
21 d |
Blood |
8 |
8 |
8 |
|
Bone marrow |
8 |
8 |
8 |
|
|
Brain |
8 |
8 |
8 |
|
We have added this information to the Appendix.
Comments 2b: The statement “A p-value of less than 0.05 (by two-tailed testing) was considered an indicator of statistical significance” is noted, but there’s no mention of correction for multiple comparisons. With numerous time points and outcome measures, this is a concern. Have you considered applying Bonferroni or false discovery rate corrections?
Response 2b: Thank you for this deep and fair comment. We would like to clarify that our statistical analysis has been differentiated according to the design of the study. In the first part of the work, where the two groups (intact and TBI) were compared at different points in time, we initially used a standard threshold of p < 0.05 for each individual comparison, considering them as independent hypotheses within the framework of a research analysis. As for the second part of the study, where three groups were compared (intact, TBI, and TBI treated with reserpine), we initially and purposefully used the Bonferroni amendment, which, unfortunately, was not explicitly stated in the text. We have corrected this omission by adding a clear mention in the methodological section (lines 211-212).
Comments 2c: For the immunohistochemical analyses, you mention “5 brain sections” and “3 fields of view”. Is this n=1 animal with technical replicates, or were multiple animals examined? This distinction is crucial for statistical validity.
Response 2c: Thank you for the question. Five brain sections were obtained from each animal. The number of cells in each section was then counted in three zones of interest (200 x 200 µm) for the cortex and three zones of interest for the SVZ.
Comments 3a: Experimental controls and groups
A reserpine-only group (without TBI) is missing. This control is essential to distinguish between reserpine’s effects on normal hematopoiesis versus TBI-induced changes. Does reserpine alter baseline bone marrow function or brain inflammation markers in healthy animals?
Response 3a: Thank you for this important clarification. In this study, we considered it unnecessary to include intact animals treated with reserpine (without TBI), as the effects of reserpine on the brain are well-known, and its effects on bone marrow function have already been demonstrated, including in our own studies. However, we will take this into account in further studies of reserpine's effects on the brain, as we suspect its effects may be more complex than currently understood.
Comments 3b: A vehicle control group is mentioned for TBI mice, but it is unclear if intact mice also received vehicle injections. This would control for the stress of repeated injections.
Response 3b: Thanks for the comment. Yes, the intact mice also received vehicle injections. We added this information in the "Materials and Methods", line 135.
Comments 3c: Have you considered using an alternative sympatholytic agent or beta-blocker as a comparison group? This would strengthen the argument that effects are specifically due to sympathetic modulation rather than off-target effects of reserpine
Response 3с: We would certainly like to expand the range of compounds being studied, and we plan to continue working in this direction. This is a pilot study, and we chose reserpine based on our experience with previous studies.
Comments 4a: An important question that deserves more discussion: To what extent can reserpine cross the blood-brain barrier, and what are its direct effects on brain-resident cells?
Reserpine can cross the BBB and directly affect central catecholamine systems. How do you distinguish between peripheral effects (bone marrow modulation) and direct CNS effects on microglia and neuroinflammation?
Response 4a: As has been correctly noted, reserpine is indeed capable of penetrating the central nervous system due to its relatively high lipophilicity. At this stage, we did not aim to separate the central and peripheral effects of reserpine. However, we hypothesize that reserpine's effects on microglia and neuroinflammation may be related to its ability to modulate signaling pathways [3-5]. We have added information about potential direct mechanisms of action of reserpine to the discussion, lines 572-586.
References:
- Verma, K.; Paliwal, S.; Sharma, S. Therapeutic Potential of Reserpine in Metabolic Syndrome: An Evidence Based Study. Pharmacol. Res. 2022, 186, 106531. doi:10.1016/j.phrs.2022.106531.
- Pallàs, M.; Vázquez, S.; Sanfeliu, C.; Galdeano, C.; Griñán-Ferré, C. Soluble Epoxide Hydrolase Inhibition to Face Neu-roinflammation in Parkinson’s Disease: A New Therapeutic Strategy. Biomolecules 2020, 10, 703. doi:10.3390/biom10050703.
- Kodani, S.D.; Morisseau, C. Role of Epoxy-Fatty Acids and Epoxide Hydrolases in the Pathology of Neuro-nflammation. Biochimie 2019, 159, 59–65. doi:10.1016/j.biochi.2019.01.020.
Comments 4b: The reduction in Iba1+ cells and microglial activation could partially result from direct reserpine effects on brain-resident immune cells rather than solely through reduced peripheral cell infiltration.
Response 4b: Thank you for pointing this out. Indeed, the reduction in Iba+ cell counts may be a consequence of reserpine's direct effect on microglia. We added information about mechanisms of reserpine's direct impact on glia (lines 572-586) in the "Discussion".
At the same time, the reduction in the number of M2 polarized microglia in the SVZ, in the absence of significant changes in the number of these cells in the cerebral cortex in the TBI region, against the background of a general reduction in inflammation, may indicate that reserpine influences microglia migration within the brain.
Comments 4c: Could you include markers that distinguish brain-resident microglia from infiltrating peripheral macrophages (e.g., TMEM119, P2RY12), though this might be challenging.
Response 4c: Thank you for your question. The Iba1 marker is a standard marker characteristic of microglia, used in most studies [6,7]. Indeed, it is possible to distinguish the population of infiltrating peripheral macrophages from resident microglia using combinations of markers, such as Iba1 and F40/80 or the aforementioned TMEM119. However, this was not the objective of this study, but we are going to explore this further.
References:
- Korzhevskii, D.E., Kirik, O.V. Brain Microglia and Microglial Markers. Neurosci Behav Physi 2016, 46, 284–290. doi: 10.1007/s11055-016-0231-z
- Franco, R., Fernández-Suárez, D. Alternatively activated microglia and macrophages in the central nervous system. Progress in neurobiology 2015, 131, 65–86. doi: 10.1016/j.pneurobio.2015.05.003
Comments 5: Line 86-87: “All manipulations with animals were maintainedin accordance with he European Convention” - “maintained in accordance with the European Convention”
Response 5: It was corrected.
Comments 6: Line 288: “TBI redused the total number of cells” - should be “reduced”.
Response 6: It was corrected.
Comments 7: The abbreviations list at the end is helpful, but CFU-G appears in the text before being defined (Added in lines 156-157).
Response 7: Thank you for your comment. It was corrected (lines156-157)
Comments 8: Figure quality: Some of the immunofluorescence images (particularly Figures 5-9) would benefit from higher magnification insets to show cellular morphology more clearly or show where the microglia are with arrows or arrow heads.
Response 8: Thank you for your comment, for a better understanding of the IGH images, we have highlighted the areas of interest in white circles.
Comments 9: The term “pathological control” is used to refer to TBI mice without treatment - consider using “TBI control” or “untreated TBI” for clarity.
Response 9: It was corrected.
Comments 10: Immunohistochemistry: You mention using “recombinant mouse monoclonal antibodies” and “rabbit polyclonal antibodies” - please specify if these are commercially available and include catalog numbers consistently (some are included, some are not)
Response 10: Thank you for your comments. We confirm that all antibodies used are commercially available. We have carefully checked and included all antibody Сatalog Numbers (Cat. No.) listed in the "Materials and Methods".
Comments 11: Line 196-198: The quantification method for IHC states “3 fields of view (200 × 200 μm area each)”. were these fields randomly selected or selected based on specific criteria? This affects potential bias
Response 11: Thank you for the question. The visual fields for the cortex in the TBI area, as well as the SVZ, were chosen randomly.
Comments 12: Statistical methods: You mention Student’s t-test for parametric data and Mann-Whitney for nonparametric, but you don’t specify how normality was assessed.
Response 12: Thank you for your comment. To assess the normality of the distribution, we used the Shapiro-Wilk test and added this information in the "Materials and methods" (lines 207-208).
Comments 13: Table 2 and Table 4: The presentation would be clearer if you bolded or highlighted significant values for easier identification
Response 13: We have highlighted the most significant values in bold in Tables 2 and 4.
Comments 14: Figure 1: The graph shows data points from 15 min to 21 days, but the scale transition from hours to days is abrupt. Consider using a log scale or breaking the axis for better visualization
Response 14: Thank you for your comment, but we think that such a presentation of the result is more illustrative.
Comments 15: Figure 2: Error bars appear very small for some measurements, please confirm these are SEM as stated and not SD. Could also justify why males?
Response 15: The error bars are indeed the SEM. Males were chosen for the experiment for two reasons. Firstly, there are more TBI cases among men than among women. And secondly, most studies use male mice or rats to model TBI.
Comments 16: Line 316-317: You state “Expression of the apoptotic marker Caspase-3 increased in Iba1+ cells of the SVZ” but the quantification in Figure 7G shows this as a percentage of Iba1+ cells, not absolute numbers. Please clarify whether total Caspase-3+ cell numbers also increased
Response 16: The total number of cells expressing Caspase-3 after TBI did increase, but we decided not to publish these data, because in this study we focused on Iba1+ cells, and in the future, it will be much more interesting to show the effect of reserpine on the background of TBI, on the expression of Caspase-3, by various brain cells such as various populations of neurons, progenitor cells, as well as neuroglial cells and oligodendrocytes.
Comments 17: Lines 433-445: The discussion of conflicting literature on TBI-induced immune changes is good, but could be strengthened by discussing potential reasons for discrepancies (species differences, injury models, severity, timing)
Response 17: Thank you for your constructive suggestion. Indeed, the conflicting information presented in the literature is interesting. However, in our view, this topic is quite broad and requires separate consideration. In this manuscript, we focused on the most relevant TBI model and timing, based on preliminary studies.
Comments 18: Lines 492-511: The mechanism discussion would benefit from a clearer model or schematic figure showing your proposed pathway: SNS activation -> bone marrow hematopoiesis -> cell mobilization -> brain infiltration -> neuroinflammation
Response 18: Thank you for the remarks. The effect of reserpine may be associated with several mechanisms. By influencing the sympathetic innervation of the bone marrow niche, reserpine reduces the mobilization of immune cells from the bone marrow. By changing the systemic balance of catecholamines, reserpine can indirectly affect peripheral immune cells (T-cells, macrophages) migrating to the site of injury [8,9,10], and potentially microglia [11]. This is because they express α- and β-adrenergic receptors, the activation of which depends on the concentration of catecholamines. This may affect the pro- or anti-inflammatory response [11]. Another potential mechanism of the anti-inflammatory action of reserpine is its inhibition of soluble epoxyhydroxylase (sEH) [12]. sEH inhibition reduces the NF-κB signaling pathway activation, leading to a decrease in the production of proinflammatory cytokines such as TNF-α [13]. In addition, the expression of inducible nitric oxide synthase (iNOS), characteristic of M1-polarized microglia, is reduced. On the other hand, a decrease in sEH activity was accompanied by an increase in the production of brain-derived neurotrophic factor (BDNF) by astrocytes and vascular endothelial growth factor (VEGF) by neurons, which have a neuroprotective and regenerative effect [14]. We added this information to the "Discussion" (Lines 572-586). Moreover, we have added a figure illustrating the potential mechanism of action of reserpine in TBI.
References:
- Gopalakrishnan, A.; Sievert, M.; Ruoho, A.E. Identification of the Substrate Binding Region of Vesicular Mon-oamine Transporter-2 (VMAT-2) Using Iodoaminoflisopolol as a Novel Photoprobe. Mol. Pharmacol. 2007, 72, 1567–1575, doi:10.1124/mol.107.034439.
- Wimalasena, K. Vesicular Monoamine Transporters: Structure-Function, Pharmacology, and Medicinal Chemistry. Med. Res. Rev. 2011, 31, 483–519, doi:10.1002/med.20187.
- Li, W.; Knowlton, D.; Woodward, W.R.; Habecker, B.A. Regulation of Noradrenergic Function by Inflammatory Cy-tokines and Depolarization. J. Neurochem. 2003, 86, 774–783.
- Liu, H.; Leak, R.K.; Hu, X. Neurotransmitter Receptors on Microglia. Stroke Vasc. Neurol. 2016, 1, 52–58, doi:10.1136/svn-2016-000012.
- Pallàs, M.; Vázquez, S.; Sanfeliu, C.; Galdeano, C.; Griñán-Ferré, C. Soluble Epoxide Hydrolase Inhibition to Face Neuroinflammation in Parkinson’s Disease: A New Therapeutic Strategy. Biomolecules 2020, 10, 703, doi:10.3390/biom10050703.
- Pallàs, M.; Vázquez, S.; Sanfeliu, C.; Galdeano, C.; Griñán-Ferré, C. Soluble Epoxide Hydrolase Inhibition to Face Neuroinflammation in Parkinson’s Disease: A New Therapeutic Strategy. Biomolecules 2020, 10, 703, doi:10.3390/biom10050703.
- Kodani, S.D.; Morisseau, C. Role of Epoxy-Fatty Acids and Epoxide Hydrolases in the Pathology of NeuroInflammation. Biochimie 2019, 159, 59–65, doi:10.1016/j.biochi.2019.01.020.
Comments 19: The discussion would benefit from addressing the clinical feasibility of reserpine for TBI, given its known side effects (depression, sedation, hypotension) and current limited use in medicine.
Response 19: Reserpine is currently used very sparingly in clinical practice due to side effects such as depression, sedation, and hypotension. In previous experiments, we observed that reducing the reserpine dose mitigates these side effects in animals. However, given the limitations of animal models, clinical use of reserpine will only be possible after clinical trials assessing its safety and efficacy in humans. We added this limitation to the "Discussion" (lines 608-609).
Comments 20: References 23 and 24 are cited in the Methods section (line 133) but appear much later in the reference list. Please ensure citations are numbered in order of appearance
Response 20: Thank you for your comment. We have made adjustments to the list of references.
Comments 21: Figure 4: The hematoxylin and eosin images would benefit from arrows or annotations indicating specific features mentioned in the text (inflammatory infiltrates, hemorrhages, vacuolization)
Response 21: Thank you for your comment, we have made changes to Figure 4, marking the specific changes with arrows.
Comments 22: Figures 5-9: Consider providing single-channel images in the supplementary materials for readers who may have difficulty interpreting merged fluorescence images.
Response 22: Thank you for your comment. We think that dual-channel and combined images will be more informative, since in a single-channel image, the lack of visualization of nuclei significantly complicates understanding.
Round 2
Reviewer 2 Report
Comments and Suggestions for Authors
Overall though, the manuscript has improved substantially, and I think it makes a solid contribution to the TBI literature. However, the remaining issues need to be acknowledged more explicitly in the limitations section.
Following are the comments addressed:
Response 2a (Sample sizes): The detailed breakdown table added to the appendix is excellent and provides complete transparency about how animals were distributed across groups and time points.
Response 2b (Multiple comparisons): Good to see they explicitly mentioned the Bonferroni correction in lines 211-212.
Response 2c (IHC quantification): The clarification helps, 5 sections per animal with 3 fields per section, and we can see from the sample table that they had n=8 animals per group.
Response 3b (Vehicle controls): They confirmed intact mice got vehicle injections too and added it to the methods.
Responses 4a & 4b (Direct CNS effects): This is actually a major improvement. The expanded discussion in lines 572-586 shows reserpine’s complexity acknowledging it can cross the BBB, discussing the sEH inhibition angle, the NF-κB pathway, and effects on catecholamine receptors.
Response 18 (Mechanistic figure): Adding a schematic figure was really needed and this should help readers a lot in understanding their working model.
Response 19 (Clinical feasibility): I appreciate that they acknowledged reserpine’s side effect profile and the limitations for clinical translation.
Minor corrections (Responses 5-13, 20-21): All the typos, figure improvements, reference ordering issues have been addressed. The addition of arrows to Figure 4 and circles to highlight areas in Figures 5-9 should make things clearer.
Response that requires additional information:
Response 1b (Alternative treatment timing): The ongoing or completed experiments with different reserpine timing protocols and processing the data for future work is great, however, it means the current paper cannot really support whether the preventive effects versus therapeutic effects or both. The authors could acknowledge that optimal treatment timing remains to be determined and will be addressed in future studies.
Responses that are somewhat needs reconsideration:
Response 3a (Reserpine-only control group): This is probably my biggest remaining concern. The authors basically said they did not think a reserpine-only group was necessary because reserpine’s effects on brain and bone marrow are well-known from previous studies, including their own work. I get where they are coming from, but I am not fully convinced by this reasoning. Sure, general effects might be documented, but that does not mean the specific effects in this experimental setup with this dose, this timing, this mouse strain, this age are necessarily the same.
Author Response
Thank you for your positive feedback on the revised manuscript. We are pleased that our clarifications and additions have adequately addressed your previous concerns.
Comments:
Response that requires additional information: Comment for Response 1b (Alternative treatment timing): The ongoing or completed experiments with different reserpine timing protocols and processing the data for future work is great, however, it means the current paper cannot really support whether the preventive effects versus therapeutic effects or both. The authors could acknowledge that optimal treatment timing remains to be determined and will be addressed in future studies.
Response to comment for responce 1b (Alternative treatment timing): Thank you for this insightful comment. We agree that the current study does not directly compare preventive versus therapeutic regimens. Determining the optimal treatment window indeed requires further investigation, and we plan to address this important question in a dedicated future study based on our preliminary data.
Responses that are somewhat needs reconsideration: Comment for response 3a (Reserpine-only control group): This is probably my biggest remaining concern. The authors basically said they did not think a reserpine-only group was necessary because reserpine’s effects on brain and bone marrow are well-known from previous studies, including their own work. I get where they are coming from, but I am not fully convinced by this reasoning. Sure, general effects might be documented, but that does not mean the specific effects in this experimental setup with this dose, this timing, this mouse strain, this age are necessarily the same.
Response to comment for responce response 3a (Reserpine-only control group): Thank you for raising this critical point. We fully acknowledge the validity of your concern. While we relied on established literature and our previous work to infer the specific effects of reserpine alone, we agree that a reserpine treated mice without TBI group within the exact parameters of this study (strain, age, dose, timing) would have provided the most rigorous evidence to isolate the drug's effect in the context of TBI.
Given that the current experimental series is completed, we are unable to add this control group retrospectively. To address this limitation transparently:
- We have added a statement to the Discussion section (Limitations, Lines 608-610), acknowledging the absence of a reserpine treated mice without TBI group as a study limitation. We recognize that subtle effects of the drug can vary depending on the animal line and the administration scheme used, and these effects cannot be accurately assessed in our particular model.
- We consider this an essential control for all future studies building on this work. The design of our next experimental series already includes a reserpine-only control group to precisely characterize its effects under the same conditions used here.
Despite this limitation, we believe that the primary comparative data between TBI mice and reserpine-treated mice remain valid and informative for evaluating the effect of reserpine on TBI outcomes.